# Bayesian Post Training Enhancement of Regression Models with Calibrated Rankings

**Kevin Tirta Wijaya**[1]**, Bing Hu**[2]**, Hans-Peter Seidel**[1]**, Wojciech Matusik**[3]**, Vahid Babaei**[145]
[1]Max Planck Institute for Informatics, [2]University of Waterloo,
[3]Massachusetts Institute of Technology,[4]Fraunhofer SCAI, [5]University of Bonn
`kevintirta.w@gmail.com, b25hu@uwaterloo.ca,`
`vahid.babaei@scai.fraunhoer.de`

## Abstract

Accurate regression models are essential for scientific discovery, yet high-quality numeric labels are scarce and expensive. In contrast, rankings (especially pairwise) are easier to obtain from domain experts or artificial intelligence judges. We introduce RANKREFINE++, a novel plug-and-play method that improves a base regressor's prediction for a query by leveraging pairwise rankings between the query and reference items with known labels. RANKREFINE++ performs a Bayesian update that combines a Gaussian likelihood from the regressor and the Bradley-Terry likelihood from the ranker. This yields a strictly log-concave posterior with a unique maximum likelihood solution and fast Newton updates. We show that prior state-of-the-art is a special case of our framework, and we identify a fundamental failure mode: Bradley-Terry likelihoods suffer from scale mismatch and curvature dominance when the number of reference items is large, which can degrade performance. From this analysis, we derive a calibration method to adjust the information originating from the expert rankings. RANKREFINE++ shows a stunning $97.65\%$ median improvement across 12 datasets over previous state-of-the-art method using a realistically-accurate ranker, and runs efficiently on a consumer-grade CPU.

## 1 Introduction

Regression underpins many scientific and engineering pipelines. In materials science and drug discovery, for example, structure-to-property predictors (i.e., through regression) can guide inverse molecular design loops by replacing expensive experiments and simulations with fast surrogates (Sanchez-Lengeling & Aspuru-Guzik, 2018; Liang et al., 2021; Allen & Tkatchenko, 2022). Yet, in many domains, collecting accurate absolute labels (e.g., molecule x is toxic at 1 nM) is slow and expensive (Wijaya et al., 2024), leading to data scarcity and limiting the accuracy of learning-based regressors.

An alternative to absolute labels is pairwise ranking (e.g., is molecule x more toxic than molecule y?). Pairwise rankings can be collected from human experts or from computational models, including general-purpose large language models (LLMs), at relatively low cost and with competitive accuracy. For humans, pairwise ranking reduces cognitive load (Routh et al., 2023) and mitigate scale-interpretation bias (Hoeijmakers et al., 2024) compared to directly predicting the absolute labels. Similarly, LLMs exhibit strong pairwise ranking ability in technical domains (Guo et al., 2023; Sun et al., 2025). This raises a central question: **can we enhance a pretrained regressor using only a handful of absolute labels plus inexpensive pairwise rankings without retraining the regressor?**

We propose RANKREFINE++, a post-training enhancement method that combines a regressor's prediction with *calibrated* expert rankings via Bayesian inference. Given a pretrained regressor and an external pairwise ranker that compares a query item to a small set of labeled references, RANKREFINE++ forms a posterior over the unknown scalar by combining (i) a Gaussian likelihood centered at the regressor's prediction, and (ii) a Bradley-Terry likelihood for pairwise rankings. We show that the posterior is strictly log-concave, yielding a unique solution and enabling fast optimization.

An additional key insight is that the default Bradley-Terry model can be mismatched to the behavior of pairwise rankers, leading to biased estimates and increased errors. RANKREFINE++ addresses this with (i) a learned temperature that calibrates the sigmoid slope of the Bradley-Terry model, and (ii) an accuracy-aware soft gate that regulates the influence of the ranker.

Specifically, our contributions are:

1. We introduce RANKREFINE++ which combines a regressor likelihood with a calibrated ranker likelihood to enhance predictions without retraining. We prove that the state-of-the-art, RankRefine (Wijaya et al., 2025), is a special case under the Gaussian assumption.
2. We show how an uncalibrated Bradley-Terry model can bias the estimates when ranker likelihood dominates, and we provide a temperature calibration with accuracy-aware soft gating mechanism to mitigate the issue.
3. Across 12 cross-domain datasets (including real-world molecular datasets), RANKRE-FINE++ significantly improves over existing post-training enhancement methods (Wijaya et al. (2025); Yan et al. (2024), and 3 other baselines) and remains effective when using imperfect LLM rankers. Specifically, RANKREFINE++ achieves $19.33\%$ median MAE reduction relying on 30 reference samples and a ranker with $65\%$ accuracy, which translates to a stunning $97.65\%$ relative improvement compared to RankRefine's $9.78\%$ median MAE reduction.

Together, these results position RANKREFINE++ as a practical and principled way to leverage readily-available pairwise information to enhance scalar regressor in data-scarce domains. Source code is available at https://github.com/ktirta/regref.

## 2 BACKGROUND

**Pairwise Comparison Models.** In pairwise (binary) comparisons, the probability of item $x_i$ is preferred to $x_j$ is often modeled as $P(x_i \succ x_j) = F(y_i - y_j)$, where $y$ are latent scores and $F$ is a cumulative distribution function (Cattelan, 2012). Classic pairwise comparison models include Thurstone-Mosteller with a Gaussian *link* function (Thurstone, 1927; Mosteller, 1951) and Bradley-Terry with a logistic *link* function (Bradley & Terry, 1952). These models provide a one-dimensional likelihood for the unknown scalar label that can be estimated using MAP or MLE estimation.

**LLM-as-a-judge.** General-purpose LLMs have demonstrated strong performance on relative comparisons across various domains (Qin et al., 2024; Wu et al., 2024; Guo et al., 2023), with evidence of better performance in pairwise comparison compared to score prediction (Zheng et al., 2023). With widely available web APIs (OpenAI, 2025; Anthropic, 2025; Google, 2025), collecting comparisons at scale is increasingly easy, making LLMs a convenient external ranker when numeric absolute labels are scarce.

**Regression Refinement using Pairwise Rankings.** Two representative approaches in this area are the Projection method (Yan et al., 2024) and RankRefine (Wijaya et al., 2025). Projection constrains the regressor's prediction to a feasible interval implied by non-contradictory pairwise outcomes. Meanwhile, RankRefine fuses the regressor's output with a rank-only estimate via inverse variance weighting. Our proposed method introduces a different paradigm: it reframes fusion as a Bayesian inference, where the regressor and ranker contribute likelihoods that jointly determine the posterior.

**Relation to Learning from Human Feedback.** Recent works on fine-tuning with human feedback (Christiano et al., 2017; Ziegler et al., 2019; Ouyang et al., 2022) also use pairwise rankings modeled via Bradley-Terry, but with a different goal. They (i) learn a global reward model and (ii) fine-tune a pretrained model accordingly. In contrast, we apply a per-query, post training prediction correction by combining the regressor's likelihood with a calibrated ranker likelihood. This produces a one-dimensional posterior without modifying the regressor's parameters.

## 3 METHOD

In this section, we describe the base formulation of RANKREFINE++ (Section 3.1), along with the analyses of its behavior (Section 3.2) and proposed modifications for improving the performance (Section 3.3). We provide the detailed proofs of the analysis in the Appendix.

### 3.1 BAYESIAN INFERENCE ON EXPERT RANKINGS TO IMPROVE REGRESSION MODELS

Let $f$ be a regressor trained on $\mathbb{C} = \{(x_j, y_j)\}_{j=1}^N$, with scalar labels $y_j \in \mathbb{R}$ and prediction $f(x_j) = \hat{y}_j^{\text{re}}$. Let $R$ be an expert pairwise ranker that returns a binary comparison: $R(x_a, x_b) = 1$ if it predicts $y_a > y_b$, and $R(x_a, x_b) = 0$ otherwise. We are given a reference set $\mathbb{D} = \{(x_i, y_i)\}_{i=1}^k$ with known $y_i$, which can be, but not always is, the regressor's training set $\mathbb{C}$. For a query $x_0$ with unknown $y_0 \in \mathbb{R}$, we collect *expert rankings* against all references: $\mathbb{G} = \{r_i = R(x_0, x_i)\}_{i=1}^k$ where $r_i \in \{0, 1\}$.

Assuming conditional independence of regressor and ranker given the true $y_0$, the Bayes' rule yields

$$\underbrace{p(y_0 \mid \hat{y}_0^{\text{re}}, \mathbb{G})}_{\text{posterior}} \propto \underbrace{p(\hat{y}_0^{\text{re}} \mid y_0)\}}_{\text{reg. likelihood}} \ \underbrace{p(\mathbb{G} \mid y_0)}_{\text{rank likelihood}} \ \underbrace{p(y_0)}_{\text{prior}} \tag{1}$$

with Gaussian regressor likelihood $p(\hat{y}_0^{\text{re}} \mid y_0) = \mathcal{N}(\hat{y}_0^{\text{re}}; y_0, \sigma_{\text{re}}^2)$ and rank likelihood

$$p(r_i = 1 \mid y_0, y_i) = s(y_0 - y_i), \qquad s(z) = 1/(1 + e^{-z}), \tag{2}$$

$$p(\mathbb{G} \mid y_0) = \prod_{i=1}^k s(y_0 - y_i)^{r_i} \left(1 - s(y_0 - y_i)\right)^{1 - r_i}, \tag{3}$$

which is the product of all pairwise comparisons in $\mathbb{G}$ under the Bradley-Terry model (Bradley & Terry, 1952).

With the two likelihoods, the *base* formulation of RANKREFINE++ enhances a regressor prediction with expert rankings using a maximum *a posteriori* (MAP) estimation (Murphy, 2022) which maximizes $y$ with the objective function

$$\mathcal{L}(y) = -\frac{1}{2\sigma_{\text{re}}^2}(\hat{y}_0^{\text{re}} - y)^2 + \sum_{i=1}^k \left[ r_i \log s(y - y_i) + (1 - r_i) \log(1 - s(y - y_i)) \right] + \log p(y). \tag{4}$$

If we assume a flat (uninformative) prior, i.e., $p(y) = 1$, RANKREFINE++ reduces to a maximum likelihood estimation (MLE) (Murphy, 2022). The objective function of the RANKREFINE++ (Equation 4) is strictly log-concave so we can obtain the maximum efficiently.

**Lemma 3.1. (Strict log-concavity.)** If $\sigma_{\text{re}}^2 > 0$ and $\log p(y_0)$ is concave (including the flat prior), then $\log p(y_0 \mid \hat{y}_0^{\text{re}}, \mathbb{G})$ is strictly concave in $y_0$. $\square$

**Corollary 3.2. (Existence and uniqueness.)** The maximum *a posteriori* (and maximum likelihood under a flat prior) estimate exists and is unique. $\square$

**Proposition 3.3. (RankRefine (Wijaya et al., 2025) is a special case of RANKREFINE++ under Gaussian assumption.)** Let $\mathcal{L}_{BT}(y)$ denote the rank-only log-likelihood and $\hat{y}_0^{\text{ra}} = \arg\max_y \mathcal{L}_{BT}(y)$. By assuming Gaussianity on the ranker likelihood (similar to Wijaya et al. (2025)), the second-order expansion of $\mathcal{L}_{BT}(y)$ around $\hat{y}_0^{\text{ra}}$ yields

$$p(\mathbb{G} \mid y) \approx \mathcal{N}(y; \hat{y}_0^{\text{ra}}, \sigma_{\text{ra}}^2), \qquad \sigma_{\text{ra}}^2 = \left[ \sum_{i=1}^k s(\hat{y}_0^{\text{ra}} - y_i)(1 - s(\hat{y}_0^{\text{ra}} - y_i)) \right]^{-1}. \tag{5}$$

Combining this with the Gaussian regressor likelihood and a flat prior gives the inverse-variance weighted (IVW) (Cochran & Carroll, 1953) estimator, i.e., RankRefine (Wijaya et al., 2025), for which the estimate $\hat{y}_0^{rr}$ is,

$$\hat{y}_0^{\text{rr}} = \frac{\hat{y}_0^{\text{re}}/\sigma_{\text{re}}^2 + \hat{y}_0^{\text{ra}}/\sigma_{\text{ra}}^2}{1/\sigma_{\text{re}}^2 + 1/\sigma_{\text{ra}}^2}. \quad \square \tag{6}$$

*Proof.* The gradient and curvature of $\mathcal{L}_{BT}(y)$ are

$$\mathcal{L}'_{BT}(y) = \sum_{i=1}^k \left(r_i - s(y - y_i)\right), \qquad \mathcal{L}''_{BT}(y) = -\sum_{i=1}^k \left(s(y - y_i)(1 - s(y - y_i))\right). \tag{7}$$

The second-order Taylor expansion of $\mathcal{L}_{BT}$ around $y = \hat{y}_0^{ra}$ is

$$\mathcal{L}_{BT}(y) \approx \mathcal{L}_{BT}(\hat{y}_0^{ra}) + \mathcal{L}'_{BT}(\hat{y}_0^{ra}) \cdot (y - \hat{y}_0^{ra}) + \tfrac{1}{2}\mathcal{L}''_{BT}(\hat{y}_0^{ra}) \cdot (y - \hat{y}_0^{ra})^2. \quad (8)$$

The first term is constant and the second term is zero, therefore,

$$\mathcal{L}_{BT}(y) \propto -\frac{1}{2}\sum_{i=1}^{k} s(\hat{y}_0^{ra} - y_i)(1 - s(\hat{y}_0^{ra} - y_i))(y - \hat{y}_0^{ra})^2. \quad (9)$$

Similar to RankRefine, we make a strong assumption of Gaussianity for the BT likelihood,

$$p(\mathbb{G} \mid y_0) \propto \exp(\mathcal{L}_{BT}(y)) = \exp\left(-\frac{1}{2}\sum_{i=1}^{k} s(\hat{y}_0^{ra} - y_i)(1 - s(\hat{y}_0^{ra} - y_i))(y - \hat{y}_0^{ra})^2\right) \equiv \mathcal{N}(\mu_{ra}, \sigma_{ra}^2), \quad (10)$$

$$\mu_{ra} = \hat{y}_0^{ra}, \qquad \sigma_{ra}^2 = \Big[\sum_{i=1}^{k} s(\hat{y}_0^{ra} - y_i)(1 - s(\hat{y}_0^{ra} - y_i))\Big]^{-1}.$$

The joint likelihood of the conditionally independent regressor and ranker is now a product of two Gaussians. The MLE, i.e., MAP estimate with a flat prior, then solves a weighted least-squares problem whose maximizer is the inverse-variance weighted average. $\square$

## 3.2 ANALYSIS OF POTENTIAL PERFORMANCE DEGRADATION OF BASE RANKREFINE++

Under the base formulation, RANKREFINE++ can degrade when the reference size $k$ and ranker accuracy $a$ exceed certain thresholds. The effect is dataset-dependent and arises from a mismatch between the Bradley-Terry model and the behavior of real-world rankers.

Separating the rank likelihood from the objective in Equation 4 and maximizing it yields the rank-only MLE $\hat{y}_0^{ra}$. If the ranker were perfectly accurate and the likelihood perfectly specified, then $\hat{y}_0^{ra} \in (y_m, y_{m+1})$, where $y_m < y_0 < y_{m+1}$ are the closest references and $m = \sum_i r_i$ is the number of references ranked below $y_0$. In practice, the sigmoid in the Bradley-Terry model induces a *soft-. vs. hard-count* mismatch: the rank-only target solves a soft-count equation and need not lie in $(y_m, y_{m+1})$.

**Lemma 3.4 (Rank-only MLE may target a pseudo ground truth.)** Let $u_i(y) = s(y - y_i)$ and $m = \sum_{i=1}^{k} r_i$. The Bradley-Terry log-likelihood $\mathcal{L}_{BT}(y)$ is strictly concave and its unique maximizer $\hat{y}_0^{ra}$ satisfies

$$\mathcal{L}'_{BT}(\hat{y}_0^{ra}) = 0 \iff \sum_{i=1}^{k} u_i(\hat{y}_0^{ra}) = m. \quad (11)$$

Because $m$ is a *hard* count while $u_i$ are sigmoid *soft* counts, the solution to $\sum_{i=1}^{k} u_i(\hat{y}_0^{ra}) = m$ is a pseudo target $\tilde{y}_0$, which can lie outside $(y_m, y_{m+1})$ even under a perfectly accurate ranker, biasing $\hat{y}_0^{ra}$. See Figure 1 for illustration. $\square$

Bias from the rank-only target can increase the overall error if the rank likelihood dominates the regressor likelihood.

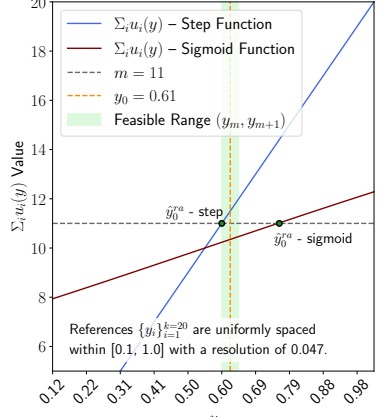

Figure 1: Soft- vs. hard-count mismatch from Lemma 3.4. We generate a synthetic reference set $\{y_i\}$ and ground truth $y_0$. From Equation 11, the rank-only MLE $\hat{y}_0^{ra}$ is the intersection between $\sum u_i(y)$ and $m$, and $(y_m, y_{m+1})$ forms the feasible range. We can see that $\sum u_i(y)$ with a step function correctly intersects the feasible range. However, using a sigmoid function results in a biased intersection outside the feasible range.

Rank likelihood dominance is governed by its curvature (Fisher information (Ly et al., 2017)), which grows with $k$.

**Lemma 3.5. (Rank curvature (Fisher information) scales with $k$.)** The Fisher information of the rank log-likelihood is $I_{\text{rank}}(y) = \sum_{i=1}^{k} u_i(y)(1 - u_i(y))$, so $I_{\text{rank}}(y)$ grows linearly with $k$. Moreover, $0 \le I_{\text{rank}}(y) \le \frac{k}{4}$. $\square$

**Lemma 3.6. (Info-weighted Newton step and dominance.)** As $k$ grows, the optimization process using Newton's method (Murphy, 2022) to solve Equation 4 is dominated by the rank likelihood term. That is, denoting $g_{\text{tot}}(y)$ and $I_{\text{tot}}(y)$ as the total gradient and Fisher information of both likelihoods at $y$, $I_{\text{reg}}$ as the Fisher information of the regressor likelihood, and $\tilde{y}^{ra}(y)$ as the target of the ranker likelihood term at $y$, a Newton step is

$$y \leftarrow y + \frac{g_{\text{tot}}(y)}{I_{\text{tot}}(y)} = \frac{I_{\text{reg}} \cdot \hat{y}_0^{\text{re}} + I_{\text{rank}}(y) \cdot \tilde{y}^{\text{ra}}(y)}{I_{\text{reg}} + I_{\text{rank}}(y)}, \tag{12}$$

which is an information-weighted average using Fisher information as the weights. □

Since $I_{\text{reg}} = 1/\sigma_{\text{re}}^2$ is constant, Lemma 3.5 implies the rank likelihood term eventually dominates as $k$ grows. Apart from Newton, for a first-order step $y \leftarrow y + \eta g_{\text{tot}}(y)$, the same decomposition yields $y \leftarrow y + \eta[I_{\text{reg}}(\hat{y}_0^{\text{re}} - y) + I_{\text{rank}}(y)(\tilde{y}^{\text{ra}}(y) - y)]$, showing similar information-weighted pull. The dominance of the rank likelihood then follows from Lemma 3.5.

Dominance alone is harmless, but combined with Lemma 3.4, a biased rank likelihood target $\tilde{y}_0$ can steer the refinement away from $y_0$. Note that dominance occurs only when both $k$ and the ranker accuracy $a$ are sufficiently large. Moreover, the accuracy threshold decreases with $k$.

**Lemma 3.7. (Accuracy-$k$ threshold for rank dominance.)** Define

$$p^* = \frac{1}{k} \sum_{i=1}^{k} \mathbb{1}(y_0 > y_i), \qquad \hat{p}(y) = \frac{1}{k} \sum_{i=1}^{k} u_i(y). \tag{13}$$

Let $a \in [0.5, 1]$ denote the ranker accuracy. Then, at any iterate $y$, the rank likelihood dominates the regressor likelihood whenever

$$a > \frac{1}{2} + \frac{|y - \hat{y}_0^{\text{re}}|}{2k\sigma_{\text{re}}^2 |p^* - \hat{p}(y)|}. \qquad \square \tag{14}$$

The right hand side of the inequality is the threshold accuracy $a_{\text{thr}}(y)$, and it decreases as $1/k$ for a fixed $y$, $\sigma_{\text{re}}^2$, and $|p^* - \hat{p}(y)| > 0$. Therefore, the required ranker accuracy such that the rank term dominates the Newton step shrinks as $k$ grows.

**Corollary 3.8. (Rank likelihood can degrade base RANKREFINE++.)** Under the base formulation of RANKREFINE++, the rank-only target can be biased (Lemma 3.4). This may lead to a performance degradation when the ranker likelihood term dominates (Lemma 3.6). In general, ranker likelihood dominance grows with the reference set size $k$ and the ranker accuracy $a$. When $k$ is large, the required $a$ for the ranker likelihood to dominate is lower (Lemma 3.7).

### 3.3 RANKREFINE++ WITH CALIBRATED EXPERT RANKINGS

The degradation stems from the Bradley-Terry modeling mismatch (Lemma 3.4): when many pairwise gaps $(y_a - y_b)$ fall on the sigmoid's transition region rather than its saturated tails, the rank-only target becomes biased. This is a *unit-scale* issue; datasets with small average $|y_a - y_b|$ have higher probability of placing many pairs on the transition slope.

We address this by introducing a temperature $\tau$ to adjust the slope, i.e., replace $u_i(y) = s(y - y_i)$ with $v_i(y; \tau) = s((y - y_i)/\tau)$. This (i) aligns the logistic slope with label units so that the tempered score $\sum_i v_i(\hat{y}_0^{\text{ra}}; \tau)$ can match the observed count $m = \sum_i r_i$, removing the *soft- vs. hard-count* mismatch, and (ii) sets the rank curvature to

$$I_{\text{rank}}(y; \tau) = \tau^{-2} \sum_i v_i(y; \tau)\big(1 - v_i(y; \tau)\big), \tag{15}$$

thereby controlling rank dominance at large $k$. We estimate $\tau$ for a dataset via a one-parameter logistic fit on the reference set, thus requiring no extra labeled data,

$$\Pr(r = 1 \mid y_a, y_b; \hat{\omega}) = s(\hat{\omega}(y_a - y_b)), \qquad (y_a, y_b) \in \mathbb{D}, \tag{16}$$

and set $\tau = \hat{\tau}_{\text{cal}} = 1/\hat{\omega}$. With this calibration of expert rankings, the Newton update is an information-weighted average with temperature-controlled curvature,

$$y \leftarrow \frac{I_{\text{reg}} \cdot \hat{y}_0^{\text{re}} + I_{\text{rank}}(y; \tau) \cdot \tilde{y}^{\text{ra}}(y; \tau)}{I_{\text{reg}} + I_{\text{rank}}(y; \tau)}. \tag{17}$$

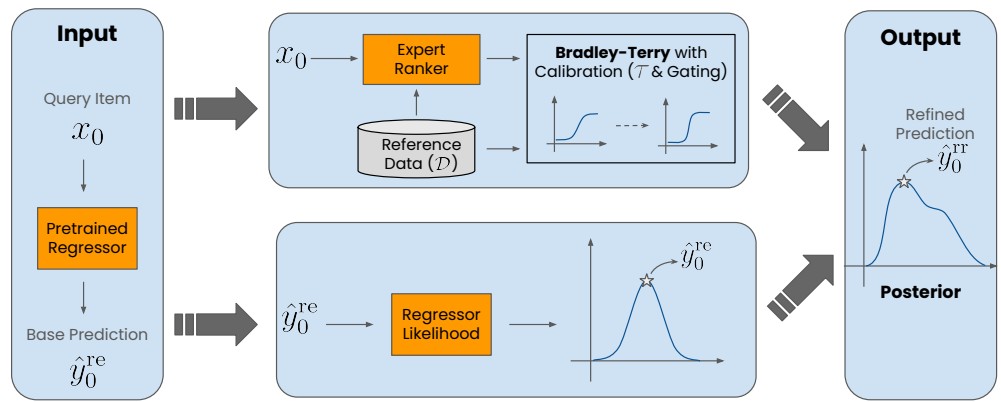

Figure 2: Illustration of RANKREFINE++. For a query $x_0$, RANKREFINE++collects the pairwise rankings from an expert ranker and converts them into a ranker likelihood via the calibrated Bradley-Terry model. The ranker likelihood is then fused with the regressor likelihood, and optionally with a prior, to define a posterior. The refined prediction $\hat{y}_0^{rr}$ maximizes the posterior distribution.

where $\tilde{y}^{\mathrm{ra}}(y;\tau) = y + \frac{\sum_{i=1}^{k}(r_i - v_i(y;\tau))}{I_{\mathrm{rank}}(y;\tau)}$.

Setting $\tau < 1$ increases the rank likelihood curvature (via the $\tau^{-2}$ factor) and shifts the update toward the rank target. When the ranker is accurate, this permits rank dominance *without* the bias highlighted by Lemma 3.4, improving performance over the naive RANKREFINE++.

On the other hand, when the ranker is less accurate, overly large curvature would let noisy rank signals dominate and harm performance. We therefore apply an *accuracy-aware soft gate*:

$$\tau(a) = 1 + (\hat{\tau}_{\mathrm{cal}} - 1)(w(a))^{\gamma}, \qquad \gamma \geq 1, \tag{18}$$

$$w(a) = \max(0, 2a - 1) \in [0, 1]. \tag{19}$$

Thus $\tau(a) \approx 1$ when $a \approx 0.5$, $\tau(a) \to \hat{\tau}_{\mathrm{cal}}$ as $a \to 1$, and intermediate $a$ produces a smooth interpolation. The final RANKREFINE++ with calibrated expert rankings is summarized in Algorithm 1, and illustrated in Figure 2.

---

**Algorithm 1** RANKREFINE++ Algorithm

---

**Inputs:** $x_0, \hat{y}_0^{\mathrm{re}}, \sigma_{\mathrm{re}}^2, \mathbb{G} = \{(x_i, y_i)\}_{i=1}^k; \mathbb{D} = \{y_i\}_{i=1}^k R;$ (optional: $p(y)$)
**Collect Comparisons:** $r_i \leftarrow R(x_0, x_i)$
**Estimate Temperature:** fit $s(\alpha(y_a - y_b))$ on labeled pairs from $\mathbb{D}$; set $\hat{\tau}_{\mathrm{cal}} = 1/\hat{\alpha}$. Estimate the ranker accuracy $a$; set $\tau \leftarrow 1 + (\hat{\tau}_{\mathrm{cal}} - 1)(\max(0, 2a - 1))^{\gamma}$.
**MAP / MLE Optimization:** Maximize Equation 4 with $s(\cdot)$ replaced by $s((\cdot)/\tau)$. With Newton steps described on Equation 17, iterate to convergence.
**Output:** $\hat{y}_0^{\mathrm{rr}}$ and approximate uncertainty $\sigma_{\mathrm{post}}^2 \approx (I_{\mathrm{reg}} + I_{\mathrm{rank}}(\hat{y}_0^{\mathrm{rr}};\tau))^{-1}$.

---

## 4 EXPERIMENTS

To demonstrate the benefits of RANKREFINE++ in data-scarce domains, we evaluate on 9 molecular property prediction datasets from the TDC ADMET regression task (Huang et al., 2021): Caco-2 (Wang et al., 2016), Clearance Microsome and Clearance Hepatocyte (Di et al., 2012), log Half-Life (Obach et al., 2008), FreeSolv (Mobley & Guthrie, 2014), Lipophilicity (Wu et al., 2018), PPBR, Solubility (Sorkun et al., 2019), and VDss (Lombardo & Jing, 2016). For each experiment, we sample $N = 100$ molecules from the original training split and $L = 100$ molecules from the original test split. We repeat this train/test resampling for 10 random seeds, similar to a Monte Carlo cross-validation. The reference set $\mathbb{D}$ is sampled from the training set with $k \in \{3, 10, 20, 30, 50, 100\}$. Two types of base regressors are used: random forest (RF) (Ho, 1995) and multilayer perceptron (MLP) (Rumelhart et al., 1986), both trained on the $N$ training labels. We also use three tabular regression datasets: crop-yield prediction from sensor data (Soundankar, 2025), student-performance prediction (Cortez, 2014), international-education cost estimation (Shamim, 2025).

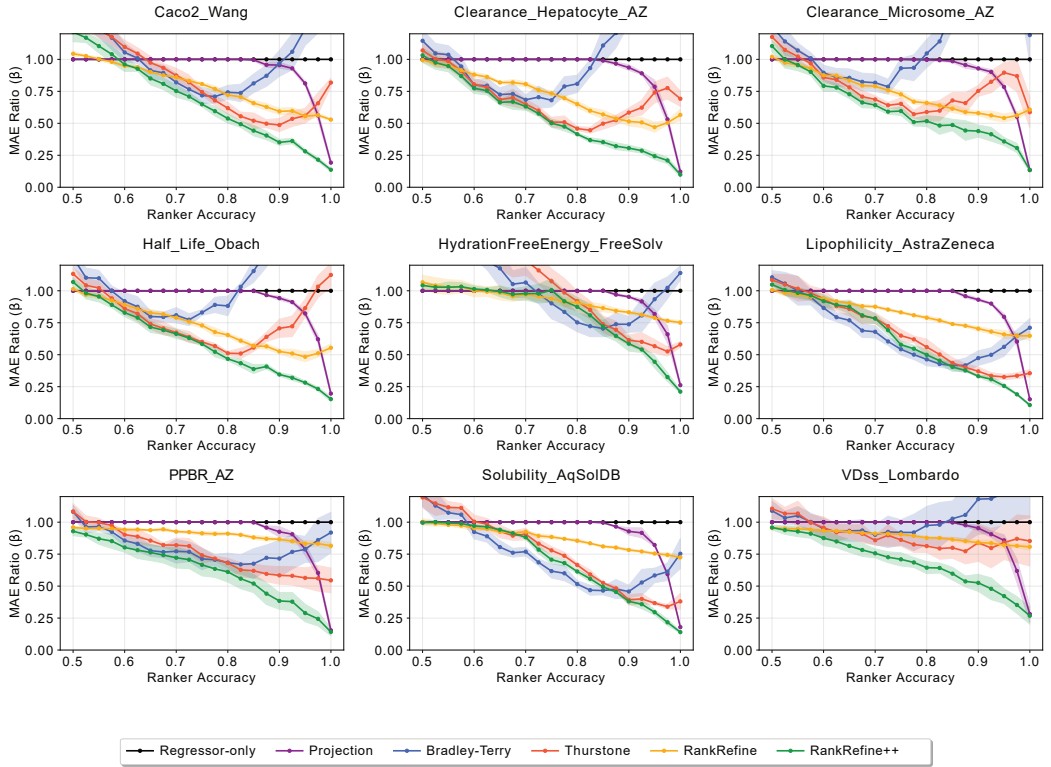

Figure 3: MAE ratio $\beta$ (lower is better) as a function of oracle ranker accuracy $a$ on nine TDC ADMET datasets ($k = 30$). We report the mean and standard deviation across 10 random splits. RANKREFINE++ (MLE-GatedTemp variant) outperforms projection and RankRefine across ranker accuracies, with especially strong gains at moderate, real-world accuracy. Results for other $k$ values with similar trends are available in Appendix.

We evaluate four RANKREFINE++ variants: MAP with a Gaussian prior, MLE, MLE with temperature scaling (MLE-Temp), and MLE with temperature scaling and soft gating (MLE-GatedTemp). The full RANKREFINE++, i.e., MLE-GatedTemp, is the default variant we use in the experiments. We compare our method to two post-training regression enhancement baselines: (i) Projection-based approach (Yan et al., 2024) and (ii) RankRefine (Wijaya et al., 2025). Additionally, we adapt two pairwise ranking models to create rank-only regression enhancement baselines: (i) Bradley-Terry (rank-only MLE using the Bradley-Terry model), (ii) Thurstone (rank-only MLE using the Thurstone-Mosteller model (Thurstone, 1927; Mosteller, 1951)). Following Wijaya et al. (2025), we report $\beta \equiv \mathrm{MAE_{post}}/\mathrm{MAE_{base}}$, which is the ratio between the post-enhancement Mean Absolute Error (MAE) and the MAE of the regressor-only predictions (lower is better).

We use two types of rankers: (i) *oracle* ranker to study the effect of ranker accuracy, and (ii) LLM ranker to demonstrate real-world use cases. For the oracle ranker, we set $r_i = \mathbb{1}(y_0, y_i)$ and then flip the outcomes with probability $1 - a$ to simulate a ranker with accuracy $a \in [0.5, 1]$. For LLM rankers, we prompt publicly-available models to compare a list of pairs of molecular text representations (SMILES (Weininger, 1988)) and measure the accuracy $a$ on a small validation set.

Additional discussions and experiments on LLM usage, temperature calibration, multi-target regression, and reference set size are available in Appendix.

## 4.1 MAIN RESULTS WITH ORACLE RANKERS

Across all nine ADMET datasets at $k = 30$, RANKREFINE++ (MLE-GatedTemp variant) consistently outperforms previous state-of-the-art, RankRefine, and 3 other baselines (Figure 3). Gains over recent custom-built enhancement methods, RankRefine and Projection, are largest in the mid-

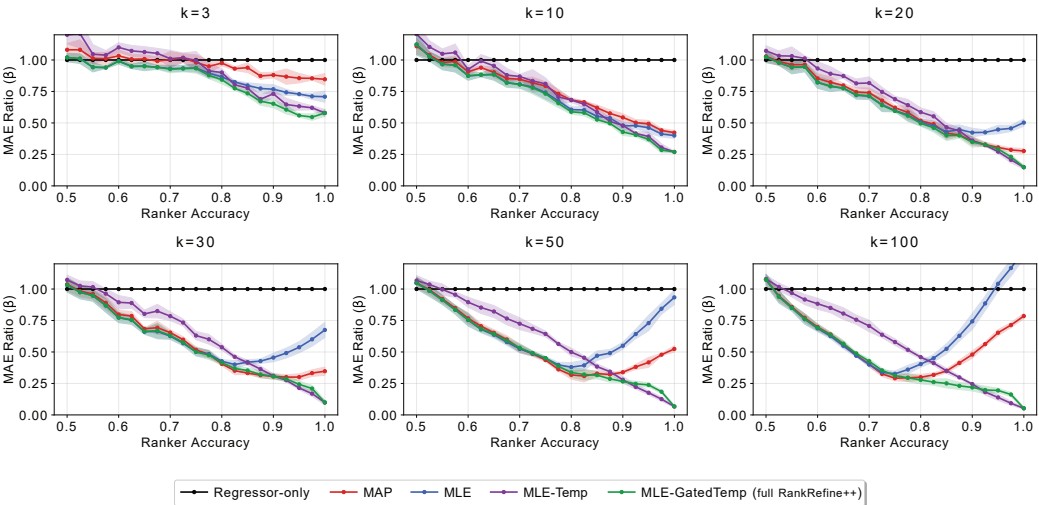

Figure 4: Effects of reference set size $k$ and temperature scaling / soft gating on the Clearance Hepatocyte dataset. The shown methods are the four variants of RANKREFINE++, with *ours* being MLE-GatedTemp. We show the MAE ratio $\beta$ (lower is better) as a function of oracle ranker accuracy $a$ for $k \in \{3, 10, 20, 30, 50, 100\}$. Without temperature scaling, MAP and MLE variants degrade at larger $k$ due to rank curvature dominance. Temperature (MLE-Temp) fixes the scale mismatch, and soft-gating (MLE-GatedTemp) further improves robustness when ranker accuracy is low.

accuracy regime (65% - 90%). At lower ranker accuracies (<65%), RANKREFINE++ matches or exceeds RankRefine and clearly outperforms Projection. At high ranker accuracies, RankRefine performance plateaus, while RANKREFINE++ continues to improve, converging to Projection's performance at perfect (yet unrealistic) ranker accuracy.

### 4.2   ABLATION: PRIOR, TEMPERATURE, AND ACCURACY-AWARE SOFT GATING

Figure 4 shows that MAP/MLE variants of RANKREFINE++ can degrade as $k$ grows, confirming that rank-dominance in the Newton step and the soft-hard count mismatch on the rank term can bias the enhancement (Corollary 3.8). The rank-only Bradley-Terry and Thurstone models in Figure 3 also exhibit similar degradation, validating that the source of degradation is the ranker likelihood. Interestingly, MAP degrades less than MLE, indicating that prior can act as a regularizer. As proposed in Lemma 3.7, the ranker accuracy threshold at which degradation appears shifts lower as $k$ increases.

Our full method (MLE-GatedTemp variant) solves the performance degradation issue. Temperature scaling aligns the sigmoid slope to the label scale and controls the rank curvature, mitigating degradation at high ranker accuracy, but can worsen performance at low ranker accuracy (Figure 4, MLE-Temp). Adding accuracy-aware soft-gating removes this trade-off (Figure 4, MLE-GatedTemp), enabling full RANKREFINE++ to deliver strong performance across the complete ranker accuracy range. Note that Figure 4 also confirms that our enhancement method works with a reference set as small as $k = 3$, which is highly practical in the real-world setting.

### 4.3   CROSS-DOMAIN GENERALITY AND REGRESSOR DIVERSITY

To test cross-domain and cross-model generalization, we add three non-molecular tabular datasets and run both Random Forest (RF) and Multilayer Perceptron (MLP) regression models. Figure 5 shows that RANKREFINE++ yields $\beta < 1$ across the ranker accuracy range, improving both RF and MLP models, and proving its generalization capability to other domains and regression models.

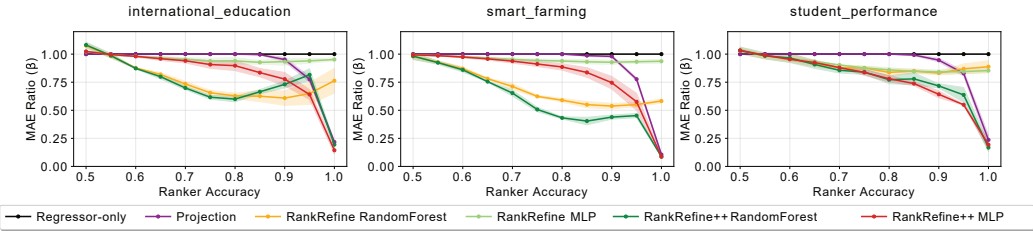

Figure 5: MAE ratio $\beta$ on three non-molecular tabular datasets with Random Forest (RF) and Multi-layer Perceptron (MLP) as base regressor models. We show that the refinements are generalizable to other domains and regressor models. Baseline (non-refined) MAEs are: **International Education Cost:** 0.0926 (MLP), 0.0609 (RF); **Smart Farming Yield:** 0.1448 (MLP), 0.1361 (RF); **Student Exam Performance:** 0.0727 (MLP), 0.0819 (RF).

| Dataset Name | Half Life | FreeSolv | PPBR | Solubility | VDss |
|---|---|---|---|---|---|
| ChatGPT5 PRA (%) | 62.20 ± 1.89 | 62.27 ± 2.00 | 63.29 ± 2.65 | 62.87 ± 3.27 | 65.95 ± 1.96 |
| RANKREFINE++ ($\beta$) | **0.952 ± 0.057** | **0.997 ± 0.021** | **0.928 ± 0.045** | **0.985 ± 0.015** | **0.853 ± 0.194** |
| RankRefine ($\beta$) | 0.977 ± 0.024 | 1.002 ± 0.022 | 0.968 ± 0.016 | 0.985 ± 0.020 | 0.952 ± 0.055 |
| Projection ($\beta$) | 1.056 ± 0.171 | 1.119 ± 0.100 | 1.007 ± 0.026 | 0.992 ± 0.039 | 0.949 ± 0.082 |
| Dataset Name | Half Life | FreeSolv | PPBR | Solubility | VDss |
| Claude4 PRA (%) | 52.36 ± 2.22 | 72.47 ± 1.40 | 51.73 ± 3.25 | 60.41 ± 2.45 | 60.85 ± 2.12 |
| RANKREFINE++ ($\beta$) | **0.955 ± 0.058** | **0.977 ± 0.011** | **0.965 ± 0.042** | 0.990 ± 0.010 | **0.850 ± 0.191** |
| RankRefine ($\beta$) | 0.981 ± 0.010 | 1.070 ± 0.052 | 0.979 ± 0.019 | **0.988 ± 0.016** | 0.951 ± 0.056 |
| Projection ($\beta$) | 1.124 ± 0.279 | 1.153 ± 0.126 | 1.031 ± 0.039 | 0.995 ± 0.014 | 0.959 ± 0.038 |

Table 1: Results using LLMs (ChatGPT5, Claude4) as external rankers, measured in $\beta$ over ten train/test splits with $N = 50, k = 20$. PRA stands for pairwise ranking accuracy. Best result for each dataset-ranker is bolded. RANKREFINE++ generally outperforms other refinement methods.

### 4.4 LLM AS IMPERFECT YET PRACTICAL RANKER

We replace the oracle ranker with off-the-shelf LLMs to obtain noisy but scalable expert rankings. With ChatGPT5 Thinking and Claude Sonnet 4 as rankers at $N = 50$ and $k = 20$, RANKREFINE++ generally achieves the best $\beta$ across five ADMET datasets. Detailed results are shown in Table 1.

### 4.5 RUNTIME ANALYSIS ON VARYING REFERENCE SET SIZE

Excluding the ranking predictions, RANKREFINE++ runs in less than 1 ms on an Intel i7-13700 CPU when the reference set size $k \leq 1000$, covering realistic scenarios in data-scarce domains. The average running time to predict expert rankings using ChatGPT5-Thinking for 1000 queries and 50 references (50,000 pairs in total) in parallel is 152.4 ± 54.7 seconds. Therefore, RANKRE-FINE++ can be run efficiently on consumer-grade computes, and LLM inference is not a significant bottleneck.

### 4.6 RANKREFINE IS A SPECIAL CASE OF RANKREFINE++.

We showed that RankRefine (Wijaya et al., 2025) is a special case of RANKREFINE++ in Proposition 3.3. We verify this empirically by comparing enhanced predictions from RankRefine and RANKREFINE++ under the Gaussianity assumptions. Across five ADMET tasks, the mean absolute difference $\Delta_y$ is at the level of machine epsilon (Table 2), confirming that Rankrefine is a special case of RANKREFINE++ with Gaussianity assumptions.

| Dataset Name | Half Life | FreeSolv | PPBR | Solubility | VDss |
|:---:|:---:|:---:|:---:|:---:|:---:|
| $\Delta_y \cdot 10^{-7}$ | $0.028 \pm 0.028$ | $0.030 \pm 5.71$ | $8.51 \pm 1.42$ | $0.029 \pm 1.33$ | $3.24 \pm 2.96$ |

Table 2: Mean absolute difference between the refined predictions of RankRefine and RANKRE-FINE++ with Gaussianity assumptions. The values are around the range of machine epsilon for a floating point precision, confirming that RankRefine is a special case of RANKREFINE++.

## 5 LIMITATIONS

While RANKREFINE++ significantly outperforms prior arts, there are several aspects that can be explored further in future works.

- **Noise model for the oracle ranker.** We inject pairwise flips uniformly to reach the desired oracle ranker accuracy. Real rankers (human or LLM) might exhibit systematic biases, which could negatively affect the Bayesian inference at lower ranker accuracies.
- **Dependence on regressor uncertainty.** RANKREFINE++ requires a base regressor that can quantify uncertainty, which many off-the-shelf regressors lack. A straightforward solution is to use ensembling or Monte Carlo dropout.
- **Reference-set selection.** Regression improvements may depend on the composition of the reference set (coverage, label diversity, resolution). Future works can explore more sophisticated, sequential sampling strategy to select the references.

## 6 CONCLUSION

We introduced RANKREFINE++, framing post training regression enhancement as a problem of Bayesian inference rather than heuristic fusion. This perspective not only generalizes prior state-of-the-art, but also reveals subtle failure modes, explains when and why performance degrades, and offers principled solutions through temperature calibration and gating. Empirically, RANKRE-FINE++ delivers consistent gains across 12 diverse datasets, operates effectively with both oracle and noisy LLM rankers, and runs under 1 ms per query. We believe RANKREFINE++ can serve as a practical bridge between label-scarce applications and the growing availability of pairwise signals from LLMs.

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

# A APPENDIX

## A.1 DETAILED PROOFS

**Lemma 3.1.** *Proof.* The regressor log-likelihood contributes $-\frac{1}{2\sigma_{re}^2}(\hat{y}_0^{re} - y_0)^2$ with curvature $-1/\sigma_{re}^2 < 0$. Each BT term adds $\log s(y_0 - y_i)$ or $\log 1 - s(y_0 - y_i)$, whose second derivative is $-s(1-s) \leq 0$. Sum of concave terms and a concave prior is concave; the Gaussian term's strictly negative curvature makes the sum strictly concave.

**Lemma 3.4.** *Proof.* $\mathcal{L}'_{BT}(y)$ (Proposition 3.3.) is strictly decreasing, therefore, $\mathcal{L}_{BT}(y)$ is strictly concave. Setting $\mathcal{L}'_{BT}(y) = 0$ gives the equation. Since $F(y) = \sum_i u_i(y)$ is continuous and strictly increasing with $F(-\infty) = 0$, $F(\infty) = k$, there is a unique solution at the observed count $m$.

Let $y_1 \leq ... \leq y_k$ be the ordered reference set and suppose the true $y_0 \in (y_m, y_{m+1})$,

- if most $(y - y_i)$ are in the sigmoid saturated regime, $F$ behaves like a hard count, so $F(y_0) \approx m$, $F(\tilde{y}_0) = m$, and $\hat{y}_0^{ra} = \tilde{y}_0 \approx y_0$ up to the resolution $(y_{m+1} - y_m)$.

- if many $(y - y_i)$ lie in the transition region, $F(y_0) \not\approx m$, $F(\tilde{y}_0) = m$, and $\hat{y}_0^{ra} = \tilde{y}_0 \not\approx y_0$. This shifts does not induce rank bias as long as $\tilde{y}_0 \in (y_m, y_{m+1})$.

- rank bias arises if the shift exits the interval, i.e., $|\tilde{y}_0 - y_0| \geq \min\{y_0 - y_m, y_{m+1} - y_0\}$, which is more likely when the number of $(y - y_i)$ that lie in the transition region is high.

**Lemma 3.6.** *Proof.* Let $I_{reg} = 1/\sigma_{re}^2$ be the Fisher information of the regressor term and write the total gradient and Fisher information at y as

$$g_{tot}(y) = g_{reg}(y) + g_{rank}(y)$$
$$= -\frac{1}{\sigma_{re}^2}(y - \hat{y}_0^{re}) + \sum_{i=1}^{k}(r_i - u_i(y)), \tag{20}$$

$$I_{tot}(y) = I_{reg} + I_{rank}(y). \tag{21}$$

Define the local rank target for a single Newton step

$$\tilde{y}^{ra}(y) = y + \frac{g_{rank}(y)}{I_{rank}(y)}$$
$$= y + \frac{\sum_{i=1}^{k}(r_i - u_i(y))}{I_{rank}(y)}. \tag{22}$$

By substituting $g_{reg}(y) = I_{reg} \cdot (\hat{y}_0^{re} - y)$ and $g_{rank}(y) = I_{rank}(y) \cdot (\tilde{y}^{ra}(y) - y)$, a single Newton step is an information-weighted average as written in Equation 12.

**Lemma 3.7.** *Proof.* Using $\mathbb{E}[r_i] = (1 - a) + (2a - 1)\mathbb{1}(y_0 > y_i)$ to model flip errors in a noisy binary ranker gives

$$\mathbb{E}[g_{rank}(y)] = \sum_{i=1}^{k}\mathbb{E}[r_i] - \sum_{i=1}^{k}u_i(y) = k\left((1 - a) + (2a - 1)p^* - \hat{p}(y)\right), \tag{23}$$

If $p^* \geq 1/2 \geq \hat{p}(y)$ or $p^* \leq 1/2 \leq \hat{p}(y)$, then

$$\left|\mathbb{E}[g_{rank}(y)]\right| \geq k(2a - 1)|p^* - \hat{p}(y)|. \tag{24}$$

Substituting $g_{rank}(y)$ with $-1/\sigma_{re}^2(y - \hat{y}_0^{re})$, the expected Newton step is rank-dominated whenever

$$(2a - 1)|p^* - \hat{p}(y)| > \frac{|y - \hat{y}_0^{re}|}{k\sigma_{re}^2}. \tag{25}$$

### A.2 Use of Large Language Models

We use large language models (LLMs) to (i) polish writing (restructuring sentences and proofread grammars and typos), (ii) search for related works, (iii) predict pairwise rankings of molecule pairs. For writing and search, we use ChatGPT5. For pairwise ranking predictions, we use ChatGPT5 and Claude Sonnet 4 (Copilot version). The prompt that we used for pairwise ranking prediction is as follow,

```
You are an expert molecular reasoning model tasked with
predicting pairwise rankings of molecules based on a described
molecular property of interest (e.g., solubility, polarity, etc.).

You will be given json files containing the test molecules to be
compared with the reference molecules. The property of interest
is described within the json files with the key "description".

* Your Task - Follow These Steps:
1. Read the dataset's description to understand the molecular
property being ranked (e.g., "higher solubility", "lower toxicity").
- Be careful with the mesurement unit. For example, a higher number
in IC50 could mean lower toxicity.
2. For each molecule pair (test molecule vs. reference molecule):
- Use your internal knowledge to infer which molecule ranks higher
for the property.
- You may use structural patterns, substrings, atom types,
SMARTS-like features, token-level patterns, or other insights
you may have.
- You are encouraged to develop your own heuristics or scoring logic
using Python.
3. Assign a "pairwise_rank" to each pair:
1 → test molecule ranks higher than reference, meaning
test_property > reference_property
0 → test molecule ranks lower or equal to reference
Caution! Only care for the value of the property. For example,
if toxicity is measured in IC50, and test_molecule IC50 value
is greater than reference_molecule IC50 value, you should output 1.
4. Save your results as a new JSON file with similiar structure.

* You Are Allowed To:
1. Write your own Python logic.
2. Use basic Python and string-based pattern recognition
3. Think step-by-step to develop useful ranking heuristics
4. Use helper functions from ChemInformatics libraries, as long
as you do not use them to directly predict the porperty of interest
values.

* You Are NOT Allowed To:
1. Use the internet to search for the property values
2. Use cheminformatics libraries like RDKit to directly predict
the property of interest values, e.g., solubility of molecule A
for solubility datasets.
3. Access files not specified in this prompt

* Each entry in your output JSON should look like this:
"test_molecule": "smiles": "...",
"reference_molecules": [
    {   "id": "...",
        "smiles": "..."
        "pairwise_ranks": 1,
```

```
    },
that is, put the pairwise_rank predictions inside the
reference_molecule. Your output json file should be named
"pairwise_ranking_predictions_{split_id}.json"

* Tips for Better Performance:
1. Think aloud: before you begin ranking, describe what is the
property of interest and why one molecule might rank higher
based on your knowledge
2. Use token or substring patterns (e.g., "more OH groups" or
"more aromatic rings")
3. Define scoring rules: e.g., "count('O') - count('N')"
to estimate polarity
```

### A.3 Effects of number of samples for temperature calibration

In Section 3.3, the temperature $\tau$ is set to $\hat{\tau}$cal, the calibrated value estimated from the labeled reference set. A natural question is the robustness of this procedure in data-scarce regimes, where only a few labeled references are available. Figure 6 shows $\hat{\tau}$cal as a function of the calibration set size $k_{\text{cal}}$. Across the 9 TDC ADMET datasets, $\hat{\tau}$cal converges at around $k_{\text{cal}} \approx 10$, indicating that optimal calibration can be obtained with only 10 labeled samples. Furthermore, except for PPBR_AZ, the estimates obtained with $k_{\text{cal}} < 10$ are already close to their converged values. This suggests that $\hat{\tau}$cal remains reliable even when very limited calibration data are available. Consistent with this observation, in the experiment where we fix the reference set size at $k = 50$ but vary $k_{\text{cal}} \in [3, 50]$ (Figure 7), the resulting performance curves are nearly identical. Together, these results demonstrate that the number of references used for temperature calibration has a negligible effect on overall regression enhancement performance.

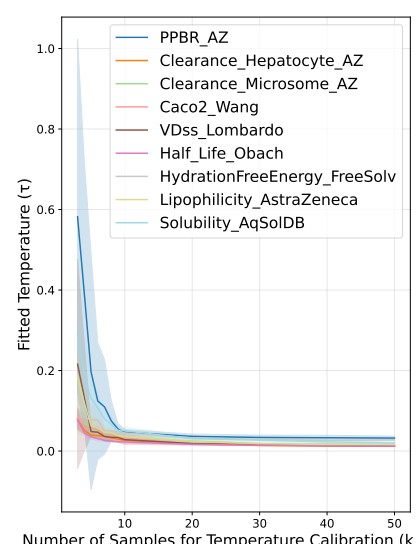

Figure 6: Effects of number of samples used to calibrate the temperature on the value of $\hat{\tau}_{\text{cal}}$.

### A.4 Regression enhancement for multiple targets

The discussion and experiments in Sections 3 and 4 focus on scalar regression tasks. In many practical settings, however, the goal is to predict multiple properties simultaneously, i.e., to produce vector-valued outputs. Extending RANKREFINE++ to this setting is straightforward: one can assume independence across output dimensions and apply RANKREFINE++ separately to each component of a target vector $\boldsymbol{y} \in \mathbb{R}^d$, yielding $d$ independent regression enhancement processes.

In practice, the components of $\boldsymbol{y}$ are often correlated. For example, in molecular chemistry, aqueous solubility and membrane permeability are often related, as are clearance and half-life. In such cases, dependencies among outputs can be captured by employing a low-rank approximation of the covariance structure, which enables a computationally efficient implementation. Exploring this extension is an interesting direction for future work.

### A.5 More Results on TDC ADMET

We show results for more $k$ values in Figure 8-Figure 16. In general, RANKREFINE++ can improve the regression error (i.e., $\beta < 1$) with a reference set size as small as $k = 3$.

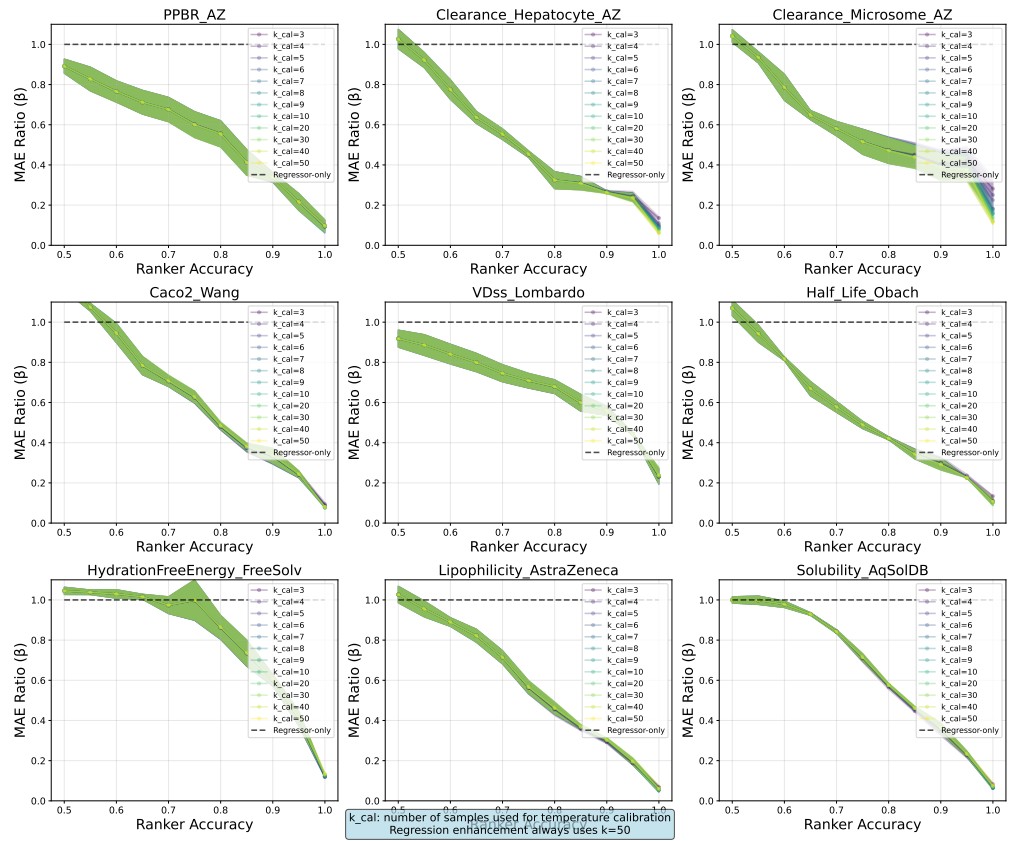

Figure 7: Effects of number of samples on the regression enhancement. We measure $\beta$ (lower is better) as a function of number of samples used to calibrate the temperature $k_{\text{cal}}$. The reference set size $k$ is set to 50.

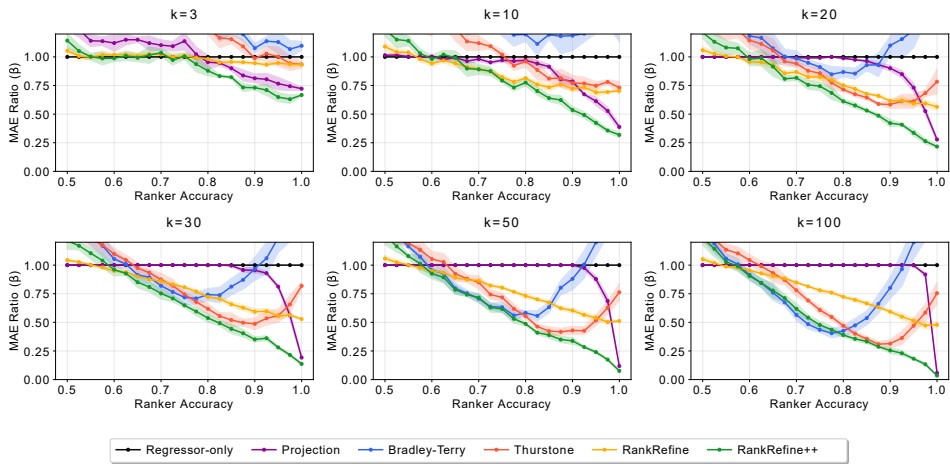

Figure 8: More results with varying reference set size ($k$) for the Caco2_Wang dataset.

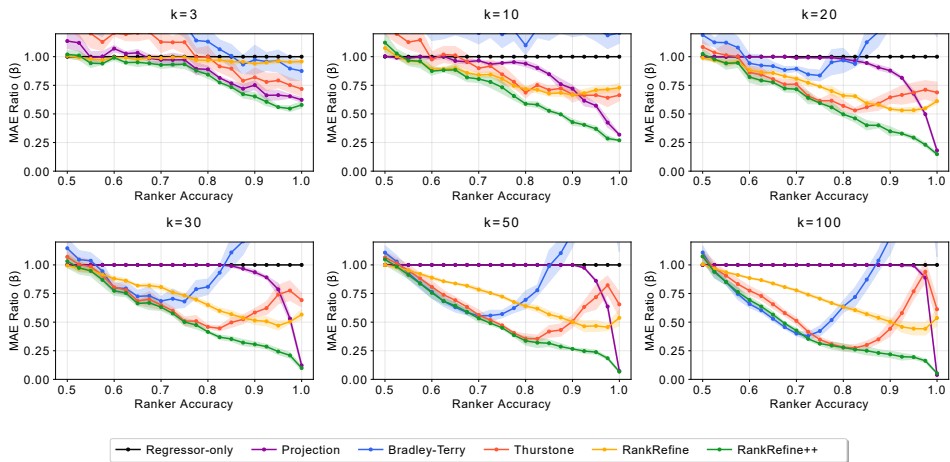

Figure 9: More results with varying reference set size ($k$) for the Clearance_Hepatocyte_AZ dataset

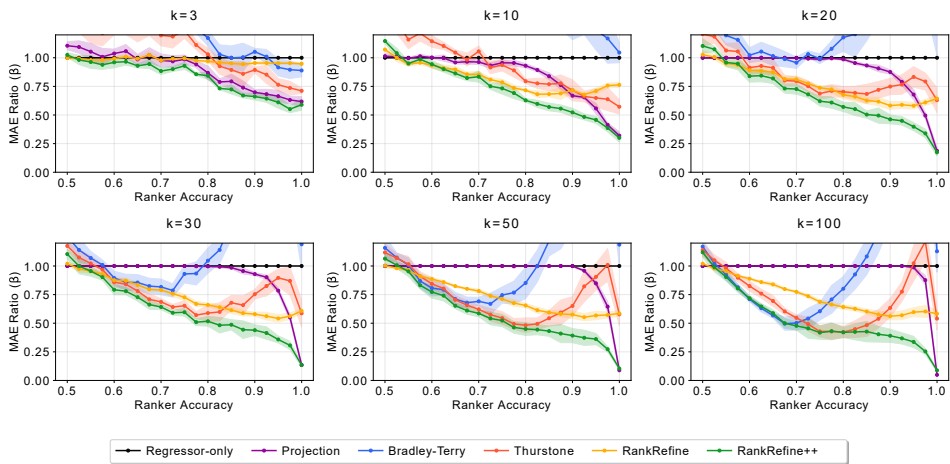

Figure 10: More results with varying reference set size ($k$) for the Clearance_Microsome_AZ dataset

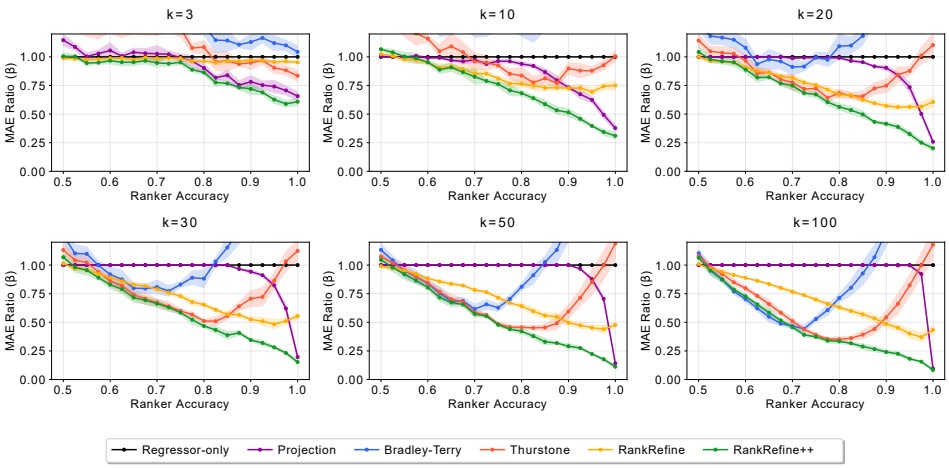

Figure 11: More results with varying reference set size ($k$) for the Half_Life_Obach dataset

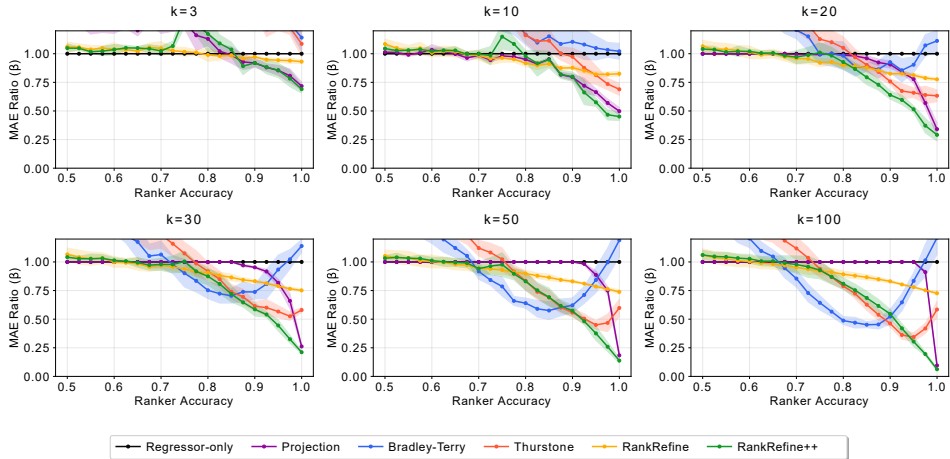

Figure 12: More results with varying reference set size ($k$) for the HydrationFreeEnergy_FreeSolv dataset

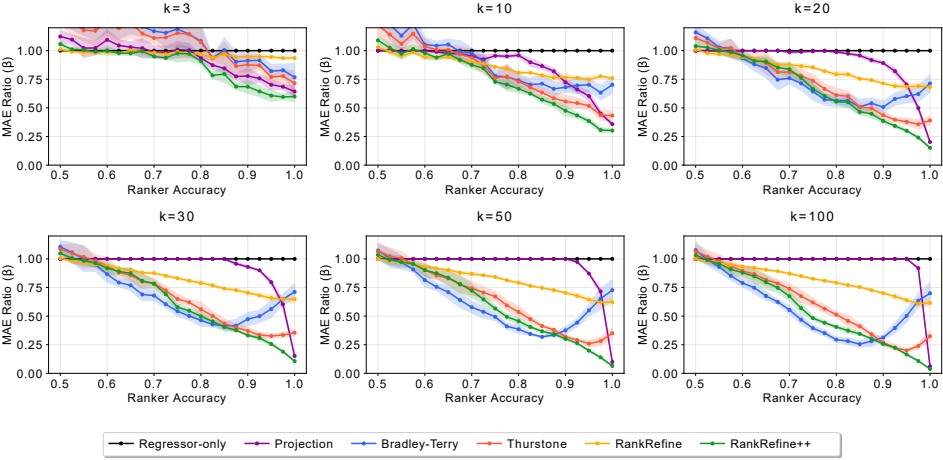

Figure 13: More results with varying reference set size ($k$) for the Lipophilicity_AstraZeneca dataset

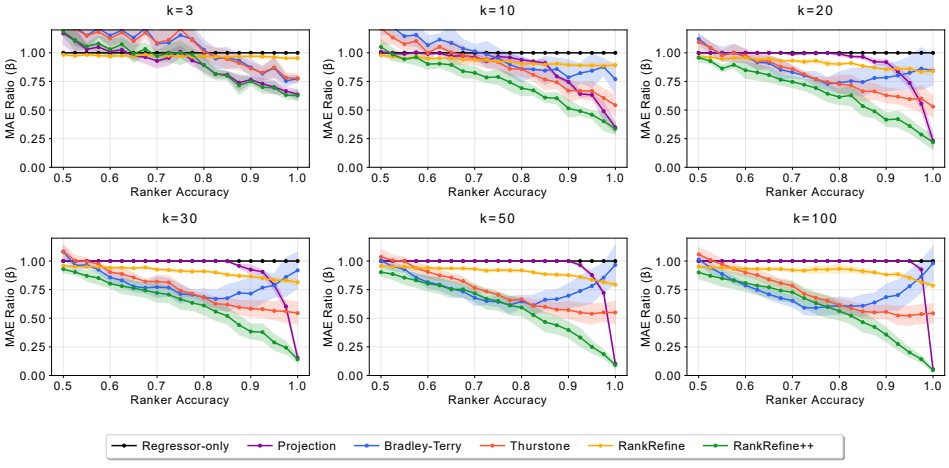

Figure 14: More results with varying reference set size ($k$) for the PPBR_AZ dataset

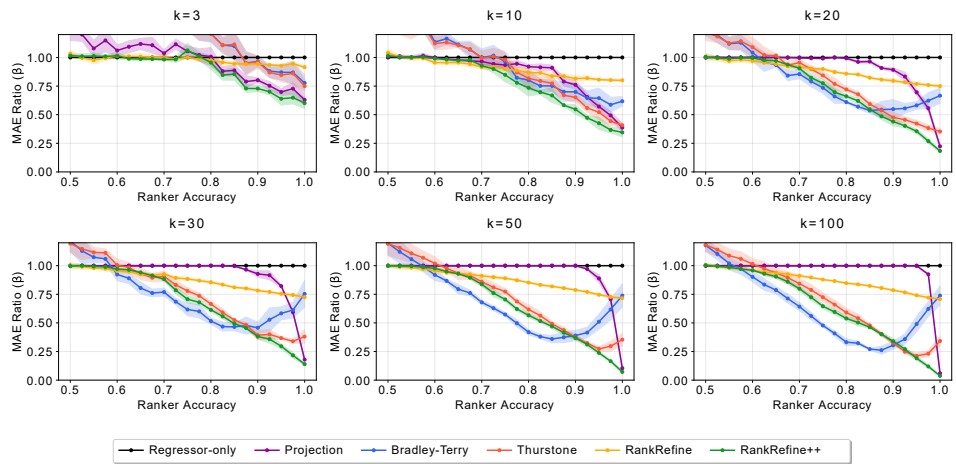

Figure 15: More results with varying reference set size ($k$) for the Solubility_AqSolDB dataset

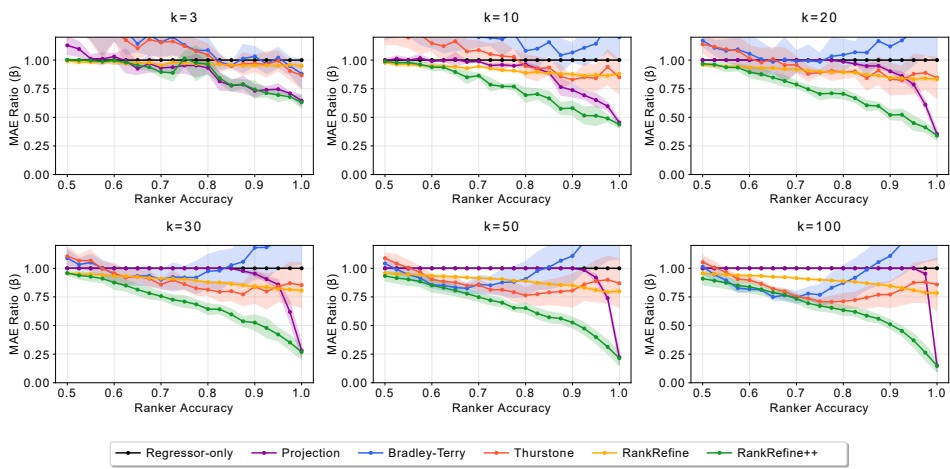

Figure 16: More results with varying reference set size ($k$) for the VDss_Lombardo dataset

### A.6 THE CORE BAYESIAN FORMULATION OF RANKREFINE++ WITHOUT CALIBRATION OUTPERFORMS PRIOR WORKS

Calibration and soft-gating offer a principled way to maximize the performance of RANKREFINE++. Nevertheless, the core Bayesian formulation of RANKREFINE++ already brings substantial improvement over the baselines by itself. As shown in Figure 17, plain RANKREFINE++ without calibration or soft-gating outperforms both RankRefine and Projection across the pairwise ranking accuracy.

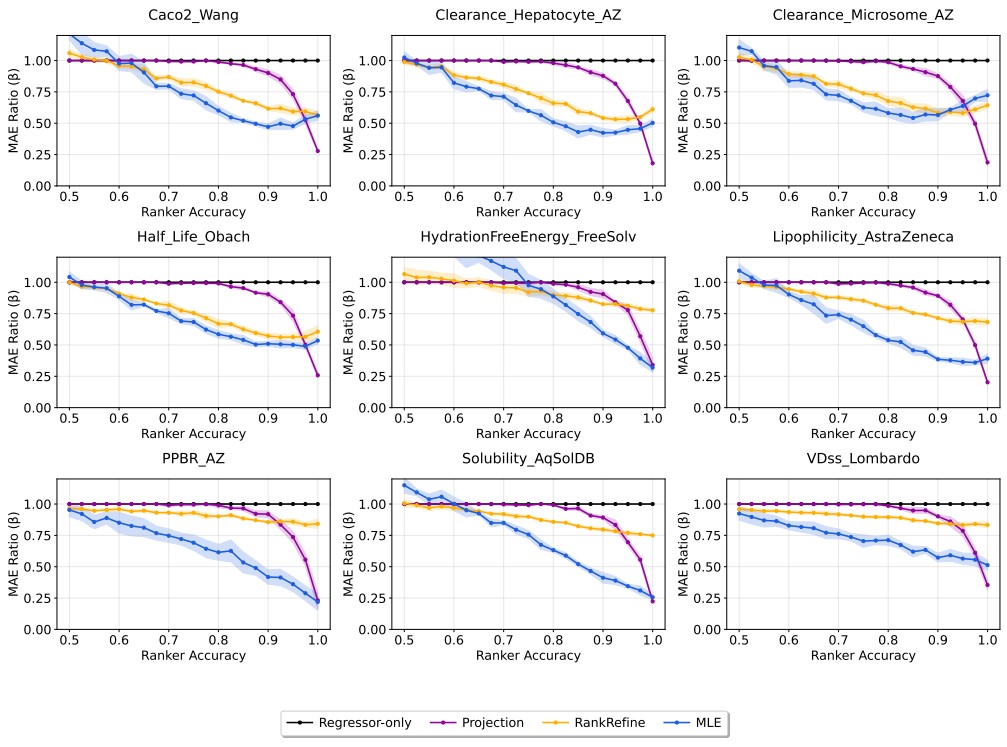

Figure 17: Plain RANKREFINE++ (MLE variant) vs. RankRefine and Projection. Reference set size k = 20. Non-calibrated RANKREFINE++ consistently achieves lower error compared to RankRefine and Projection baselines across datasets, often by a large margin

### A.7 SENSITIVITY ANALYSIS FOR UNDER/OVERESTIMATION OF REGRESSOR UNCERTAINTY

Figure 18 shows the performance of RANKREFINE++ when the pointwise uncertainty of the regressor is set to the pointwise absolute or squared error, scaled by $s$. Overestimation (e.g., $s = 10.0$) has minimal impact on the performance, while underestimation brings substantial degradation due to excessive confidence in the imperfect regressor.

### A.8 SENSITITVITY ANALYSIS FOR NOISY ESTIMATION OF THE RANKER ACCURACY

The soft-gating mechanism relies on the estimation of the ranker accuracy. Figure 19 shows the effect of noisy estimation of the ranker accuracy. When injecting the ground-truth ranker accuracy ($a \in [0.0, 1.0]$) with a Gaussian noise, there is no substantial performance degradation up to $\sigma^2 = 0.50$. Furthermore, for $\sigma^2 \geq 0.50$, the performance starts to slightly degrade only at relatively high ranker accuracy, demonstrating the robustness of RANKREFINE++ in most real-world scenarios.

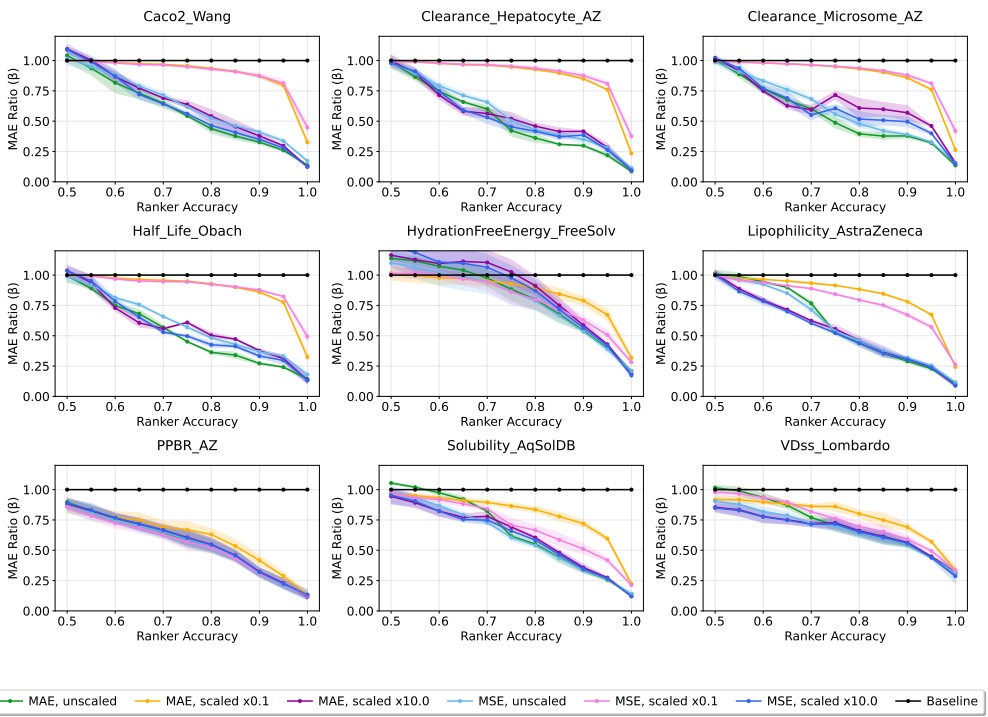

Figure 18: Sensitivity analysis on under or overestimation of regressor uncertainty. Overestimating uncertainty (e.g., s = 10.0) has minimal impact on performance, while underestimating it (e.g., s = 0.1) degrades performance due to excessive confidence in the regressor.

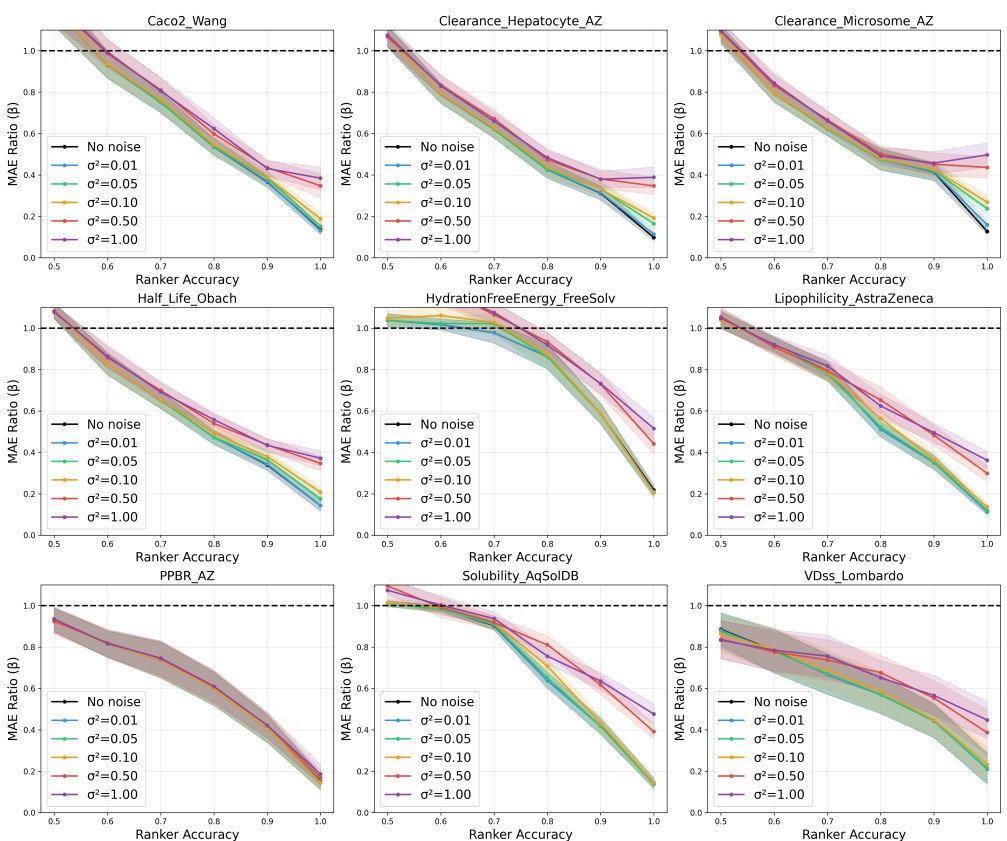

Figure 19: Sensitivity analysis when the estimation of the ranker accuracy is noisy, resulting in non-optimal $\tau$ values. RANKREFINE++ is robust even when using a suboptimal tau value caused by softgating bias

