# OpenReview forum: "Bayesian Post Training Enhancement of Regression Models with Calibrated Rankings"
_ICLR.cc/2026/Conference — ICLR 2026 Poster_

### Official Review · Reviewer_aXK6 · 2025-10-26

**Soundness:** 3
**Presentation:** 3
**Contribution:** 3
**Rating:** 4
**Confidence:** 5

**Summary:**

This paper presents a plug-and-play Bayesian enhancement to regression models. Specifically, pairwise ranking information complements scarce absolute labels as in RankRefine, but with Bayesian aggregation.

**Strengths:**

* Performance gain: Similar ideas as RankRefine but shows performance
* Efficiency: Plug-and-play avoids costly model retraining.
* Data Efficiency: Complementary pairwise signal helps data-scarce fields such as science.

**Weaknesses:**

Some questions left unanswered.
* Motivation behind pairwise: Would listwise ranking be a stronger signals?
* Generalization: While we understand science is a data-scarece domain, evaluating only on this scenario makes us wonder whether it would generalize to general regression problems.

**Questions:**

Any additional analysis to answer questions in the weakness section would help rebuttal.

---

> ### Author Response · Authors · 2025-11-21
>
> We thank the reviewer for their insightful feedback. We have conducted additional experiments to address the reviewer’s questions. The results are presented in Appendix A6, which we reference in our detailed answers below.
>
> **Weakness 1 (W1). Why pairwise ranking? Would listwise be stronger?**
>
> We agree with the reviewer that listwise ranking can offer stronger signals than pairwise ranking. However, we find that pairwise ranking is more practical and robust in our experimental setting for several reasons.
> First, given the important implications of our method for human-in-the-loop scenarios , it is well known [1] that listwise ranking increases the cognitive load on human rankers, leading to increased fatigue and a higher susceptibility to ranking errors. In addition, studies have shown that inter-ranker agreement is higher for pairwise comparisons than for listwise ranking, suggesting better data quality and reliability with the pairwise approach [2].
>
> Second, for LLM-based ranking, which is central to our work, the performance of listwise ranking can be considerably worse than pairwise. For instance, using the TDC ADME Solubility dataset with ChatGPT5, we observed the following performance difference in Pairwise Agreement Ratio (comparing the predicted to the ground-truth rankings in a pairwise manner):
> - Pairwise (binary): 62.87% ± 3.27%
> - Listwise (trinary): 58.32% ± 4.29%
>
> The Pairwise Agreement Ratio for listwise ranking is obtained by decomposing the listwise ordering into pairwise ordering.
>
> Third, we take note of the potential for extending the Bayes-ECR framework to incorporate listwise rankings. This could be achieved by replacing the Bradley-Terry model with the Plackett-Luce model for calculating the listwise likelihood. We reserve this as a direction for future work.
>
> **W2. Generalization beyond molecular science.**
>
> We recognize that generalization to other domains is a critical characteristic. In Section 4.3, we evaluated Bayes-ECR across three non-chemistry domains, i.e., international education cost, farming yield, and student exam performance, demonstrating its effectiveness in diverse domains. Moreover, thanks to the feedback, we are including additional results (**Fig. 19 in Appendix A6**) showing strong performance even when training data is not scarce (e.g., with 1000 training labels), indicating our method's robustness across different data-size regimes (and therefore different applications).
>
> **References**
>
> [1] Fürnkranz, J., Hüllermeier, E. (2010). Preference Learning and Ranking by Pairwise Comparison. In: Fürnkranz, J., Hüllermeier, E. (eds) Preference Learning. Springer, Berlin, Heidelberg. https://doi.org/10.1007/978-3-642-14125-6_4
>
> [2] Tarlow, Kevin R., et al. "Reliable visual analysis of single-case data: A comparison of rating, ranking, and pairwise methods." Cogent Psychology 8.1 (2021): 1911076.

---

> > ### Comment · Reviewer_aXK6 · 2025-11-26
> >
> > Thanks for the response, your response to W2 makes sense, but regarding listwise, citations would not address my concerns-- there are many papers claiming otherwise (listwise aligning better), and cognitive load of listwise has a trade-off. For example, for ranking k elements, listwise ranking on k incurs higher load than pairwise comparison, but it requires C(k,2) pairwise comparisons to collect comparable information.

---

> > > ### Author Response · Authors · 2025-11-26
> > >
> > > We highly appreciate the reviewer’s timely engagement. Regarding the cognitive load trade-off, we believe that there is a slight misunderstanding of our problem setting which we would like to clarify precisely. We are also happy to provide a proof of concept of our method extending to listwise rankings.
> > >
> > > **Number of pairwise rankings to collect comparable information to listwise ranking.**
> > >
> > > The reviewer is correct that, in the general case, recovering comparable information as listwise ranking over k+1 items requires C(k+1,2) pairwise rankings. However, our setup involves **only one** unlabeled query item and k reference items whose ground-truth labels (and thus, the ordering) are already known. We only need to determine how the query item ranks relative to the k reference items. As a result, only k pairwise rankings, $(\text{query}, \text{re}f_i)_{i=1}^{k}$ are relevant, and *not* all C(k+1,2) pairs.
> > >
> > > Therefore, for our specific setup, listwise ranking does not reduce the number of necessary comparisons below k, but retains the disadvantage of requiring higher cognitive load over pairwise rankings.
> > >
> > >
> > > **Extension to listwise ranking.**
> > >
> > > As we mentioned previously, our framework can be extended to listwise ranking by using a Plackett-Luce likelihood. We implemented a prototype of this extension, using the same noisy pairwise oracle ranker as in the main experiments to ensure a fair comparison with other methods.
> > >
> > > Let us define a query $x_0$ with unknown scalar label $y_0$, a reference set with known labels $D = {(x_i, y_i)}$, and pairwise rankings $G = [r_i = 1(y_0 > y_i)]$. The Plackett-Luce model,
> > > $$\text{Pr}(\pi_t = i | S_t) = \frac{exp(y_i)}{\sum_{j\in S_t}exp(y_j)},$$
> > > gives the probability that item i is chosen at sampling step t from the candidate set $S_t$, where $S_1 = {0, ..., k}$ contains the query and all references, and $S_{t+1} = S_t \setminus \{\pi_t\}.$
> > >
> > > We construct a full listwise ranking as follows. First, sort the references by their known labels in ascending order,
> > > $$y_(1) \leq ... \leq y_(k).$$
> > >
> > > Let $m = \sum_i r_i$ be the number of wins of the query against the references. We then insert the query at position $r = m+1$ in the sorted list, yielding a permutation $\pi$ over {0, 1, ..., k}. The listwise likelihood of the permutation $\pi$ is then
> > > $$p(\pi | y_0) = \prod_{t=1}^{k+1}\frac{exp(y_{\pi_t}}{\sum_{j \in S_t} exp(y_j)}.$$
> > >
> > > If the query appears at position $r$ in $\pi$, only the first $r$ sampling steps depend on $y_0$. Writing
> > > $$B_t = \sum_{j \in S_t \setminus \{0\}} exp(y_j),$$
> > > the negative log likelihood involving $y_0$ is,
> > >
> > > $$L(y_0) = \sum_{t=1}^r [log (B_t + exp(y_0)) - y_0].$$
> > >
> > > With this listwise formulation, we obtain **new** results reported in **Fig. 20 Appendix A6**. For $k = 3$, the listwise variant achieves reasonable performance, confirming that Bayes-ECR can indeed be extended to listwise ranking. However, its performance is not yet on par with our pairwise version. Two likely major contributing factors are:
> > > - In Plackett-Luce, each step normalizes by a sum of $exp$ over $S_t$, which can amplify noise from references far from the query, and
> > > - Our insertion rule $r = m + 1$ is somewhat brittle, but is required here to keep the oracle ranker’s effective accuracy comparable across pairwise and listwise settings.
> > >
> > > We believe that listwise rankings have the potential to provide stronger signals than pairwise rankings, but they are more sensitive to each individual component (e.g., ranker’s accuracy, likelihood model). We will discuss these implications in the paper and look forward to designing a more robust listwise approach  in the future work.

---

### Official Review · Reviewer_JneQ · 2025-10-31

**Soundness:** 3
**Presentation:** 3
**Contribution:** 2
**Rating:** 4
**Confidence:** 4

**Summary:**

This paper addresses the challenge of improving regression models in data-scarce domains, where high-quality numeric labels are expensive, but pairwise rankings are relatively cheap to obtain. The authors propose Bayesian Enhancement with Calibrated Ranking (BAYES-ECR), a novel and efficient "plug-and-play" method to enhance a pre-trained regressor's predictions post-training, without requiring any retraining. The core idea is to perform a per-query Bayesian update. For a given query, BAYES-ECR combines two sources of information: (1) A Gaussian likelihood from the pre-trained regressor, centered on its prediction $\hat{y}_{0}^{re}$ with variance $\sigma_{re}^{2}$. (2) A ranker likelihood derived from pairwise comparisons between the query and a small set of $k$ reference items with known labels. This likelihood is modeled using the Bradley-Terry model.

**Strengths:**

The main contribution of the work is its analysis of where the Bradley–Terry model breaks down. Although using BT models for ranking is fairly common, the authors offer a fresh insight: the “soft-count” sigmoid mechanism interacts with the number of reference points, $k$, in a way that creates a kind of curvature dominance—an effect that amplifies bias. This observation is both new and important. This insight moves the field from heuristic fusion (like inverse-variance weighting) to a more principled understanding.

**Weaknesses:**

1. The framework builds on combining a regressor likelihood, $\mathcal{N}(\hat{y}{0}^{re}; y_0, \sigma{re}^2)$, with a ranker likelihood. The Newton step works as an information-weighted average, where the regressor’s contribution is scaled by $I_{reg} = 1/\sigma_{re}^2$. This makes $\sigma_{re}^2$ a key parameter. The paper notes that many regressors don’t naturally provide this value and suggests using ensembling or dropout to approximate it—but it never clearly explains how $\sigma_{re}^2$ was actually estimated for the Random Forest and MLP models in the experiments. Was it treated as a single hyperparameter tuned on a validation set?

2. The accuracy of the ranker, $a$, is necessary to operate the accuracy-aware soft gate, $\tau(a)$. According to the authors, a "small validation set" is used to gauge this accuracy. However, rather than focusing on how sensitive the results are to the accuracy estimate itself, the robustness checks in the appendix examine how well $\hat{\tau}_{cal}$ is calibrated (see Figures 5 and 6). The estimate of $a$ may not be accurate if that validation set is very small, which could render $\tau$ less than ideal. The robustness claims would be much stronger if a brief sensitivity analysis of how $\tau$ varies with various estimates of $a$ were included.

**Questions:**

1. Could the authors please clarify precisely how the regressor variance $\sigma_{re}^2$ was estimated for the Random Forest and MLP experiments? Given its critical role as $1/I_{reg}$ in the information-weighted update step, this detail is essential for reproducibility and for understanding the method's behavior. Did you, for example, estimate $\sigma_{re}^2$ as the Mean Squared Error of the base regressor on a hold-out set?

2. According to Section 3.3, labeled pairs from the reference set $\mathbb{D}$ are fitted with $s(\hat{\omega}(y_a - y_b))$ to estimate $\hat{\tau}_{cal}$. Is the true rankings (i.e., $r_i = 1$ if $y_a > y_b$) obtained from the ground-truth labels in $\mathbb{D}$ used in this fitting process? Or does it make use of the external ranker's noisy rankings $r_i = R(x_a, x_b)$?

3.

---

> ### Author Response · Authors · 2025-11-21
>
> We thank the reviewer for their insightful feedback. We have conducted additional experiments to address the reviewer’s questions. The results are presented in Appendix A6, which we reference in our detailed answers below.
>
> **Weakness 1 (W1) / Question 1 (Q1). How uncertainty $\sigma_\text{re}^2$ is estimated for Random Forest and MLP?**
>
> We thank the reviewer for pointing out this missing detail. In our implementation, $\sigma_\text{re}^2$ is the quantified uncertainty of the point estimate, that is, each query will have its own uncertainty estimation. As such, we did not estimate $\sigma_\text{re}^2$ based on a hold out set, but instead directly used the variance of the Random Forest model, or Monte Carlo dropout for MLP. We will update the experimental setup in Section 4 to clarify this approach.
>
> **W2. The estimate of ranker accuracy (a)  may not be accurate if the validation set is very small, which could render temperature ($\tau$) less than ideal.**
>
> The reviewer has a crucial observation: a very small validation set could indeed lead to an inaccurate estimate of $a$, potentially resulting in a suboptimal $\tau$ value that could affect the regression improvement ($\beta$) metric. However, in Figure 5 in Appendix, we showed that the value of $\tau$ stabilizes after the validation set reaches just 10 samples, suggesting that a useful estimate of $\tau$ can be achieved with a reasonably (but not too) small number of samples.
> Furthermore, in a new experiment, we investigated the effect of noise in the ranker accuracy estimate on $\beta$. We injected Gaussian noise into the ranker accuracy estimate used to determine $\tau$ according to the following formula:
>
> $$\text{noisy}_a = \text{max}(\text{min}(a + N(0, \sigma^2), 1.0), 0.5),$$
>
> where $\sigma^2$ is chosen from $\{0.01, 0.05, 0.10, 0.50, 1.00\}$.
> The results of this experiment (**Fig. 18 in Appendix A6**) reveal two key points:
> 1. A substantial drop in improvement is primarily observed when the ranker accuracy is high (80% or more). In the lower ranker accuracy range, the performance degradation is not significant.
> 2. Even with this performance degradation, the resulting curves still outperform RankRefine. This suggests that, at a system level, Bayes-ECR is robust even when using a suboptimal $\tau$ value caused by softgating bias.
>
> **Q2. Does the fitting process to estimate $\tau$ use true rankings or external ranker’s noisy rankings?**
>
> We use the true rankings derived from the ground-truth labels in $\mathbb{D}$ for this fitting process, rather than the external ranker's noisy rankings.

---

### Official Review · Reviewer_zBQs · 2025-10-31

**Soundness:** 3
**Presentation:** 3
**Contribution:** 3
**Rating:** 8
**Confidence:** 4

**Summary:**

In this work, authors propose BAYES-ECR, a novel plug-and-play post-training method designed to improve the predictions of a base regression model. The key idea is to leverage pairwise rankings between a query item and a set of reference items (with known labels), often easier and cheaper to obtain than high-quality numeric labels. BAYES-ECR is based on a Bayesian update using Gaussian likelihood (from base regressor) and another likelihood from external pairwise ranker. Authors find that the resulting posterior is log-concave thereby ensuring a unique maximum likelihood solution which can be recovered using efficient optimization procedures like Newton updates.

**Strengths:**

I find the following strengths in this paper:
- I find combining regressor predictions with pairwise ranking information is a novel way as the form of Bayesian update, a cool idea and which I believe is this work’s significant conceptual strength, moving beyond other heuristic approaches.
- The core problem, i.e. scarcity of high-quality numeric labels versus the availability of easier-to-obtain pairwise rankings, is highly relevant in many scientific and engineering domains (not just LLMs), for e.g. such as materials science and drug discovery, therefore this method has a wide applicability.
- I find the proposed temperature calibration and accuracy-aware soft gating mechanisms to be well-motivated by the theoretical analysis.
- Based on the empirical findings, the proposed method runs very efficiently and from what I find, inference is unlikely to be a bottleneck for this method.
- The study by including off-the-shelf LLMs as imperfect rankers highlights the method's applicability to real-world scenarios.

**Weaknesses:**

Few weaknesses that I would like the authors to address:
- The core Bayesian update relies on the assumption of conditional independence between the regressor and ranker - while I think this assumption may make some sense in general, but I also think thai may not always perfectly hold in practice, especially if the ranker's "expert" knowledge is implicitly derived from data similar to the regressor's training data. Is this something the authors have considered or have any view on? How will the method change if this is the case?
- The paper mentions that regression improvements may depend on the composition of the reference set and suggests future work on sophisticated sampling strategies. A brief discussion of potential biases / optimal strategies for reference set selection, even if speculative, could enrich the overall study in this paper.
- One of the commonly discussed limitations of the Bradley-Terry model is that it inherently assumes transitive relationship among the pairwise comparisons, i.e. if A > B and B > C, then A > C. While this is usually fine in many cases, some real-world "expert" rankings might be locally inconsistent or reflect non-transitive preferences in certain contexts, which could challenge the model. Can the authors comment on such a setup? Or what impact it can have on such applications? Finally, any thoughts on how the current framework can be extended to handle it?

**Questions:**

Some additions questions to the authors:
- How sensitive is BAYES-ECR's performance to inaccuracies or miscalibration of the base regressor's uncertainty estimate?
- How much is the computational burden of the Newton update steps in real-world applications? Can the authors provide some clear numbers to understand the overhead for using the proposed method?

---

> ### Author Response · Authors · 2025-11-21
>
> We thank the reviewer for their insightful feedback. We have conducted additional experiments to address the reviewer’s questions. The results are presented in Appendix A6, which we reference in our detailed answers below.
>
> **Weakness 1 (W1). What if the ranker and regressor are correlated?**
>
> We appreciate this thoughtful point. In practice, we believe that Bayes-ECR remains effective even when the regressor and ranker share information sources. The baseline method, RankRefine [1], already showed measurable gains when both regression values and pairwise rankings came from the same human experts, suggesting that strict conditional independence is not essential for gaining practical improvement.
>
> Moreover, Bayes-ECR’s Bayesian formulation naturally extends to model such dependencies. Let $\epsilon_\text{rank}$ be the error term of the ranker, $\epsilon_\text{reg}$ the error term of the regressor, $r_i​ = \mathbb{1}((y_0​−y_i​) + \epsilon_\text{rank} ​> 0)$, and $y_0^\text{​reg} ​= y_0​+\epsilon_\text{reg}$​,
> and let $\text{Cov}(\epsilon_\text{rank}​, \epsilon_\text{reg}​) = \rho \sigma_\text{rank} \sigma_\text{reg}​$.
> Assuming that both error terms are zero-mean normals,
> $\mathbb{E}[\epsilon_\text{rank} \vert \epsilon_\text{reg}] = \rho (\sigma_\text{rank} / \sigma_\text{reg}) (y_0^\text{reg} - y_0))$.
> Thus, the rank likelihood can be corrected as $s((y_0​−y_i​) + c(y_0^\text{​reg}​−y_0​))$, where  $c \propto \rho(\sigma_\text{rank}​\/\sigma_\text{reg}​)$.
>
> Intuitively, when the regressor overestimates ($y_0^\text{reg}>y_0$) and the errors are positively correlated  ($\rho>0$), the ranker tends to rank $y_0$ too high; this coupling term captures and compensates for that shared bias.
>
> **W2. Potential sampling strategy.**
>
> In Section 5 of the main text, we hypothesize that Bayes-ECR benefits most when the reference set achieves (i) **good coverage** near the query, (ii) **high diversity**, and (iii) **high resolution** in reference values (i.e., in y, not x). A simple strategy to improve (i) is to select the k nearest neighbors of the query in feature space, under the assumption that proximity in features correlates with proximity in labels. One can also introduce a diversity regularizer that penalizes overly similar label values among the selected references, thereby balancing local coverage and label diversity in the reference set, improving both (i) and (ii). If we have access to an acquisition function (e.g., an experimental lab), we can also integrate more sophisticated methods such as Bayesian optimization to obtain a reference set that is optimal in coverage, diversity, and resolution.
>
> **W3. What if the ranker is locally inconsistent and the rankings have non-perfect transitivity?**
>
> This non-perfect transitivity scenario has naturally been covered by our experiments. Since perfect transitivity is only achievable with an ideal ranker of 100% accuracy, our experiments with oracle-based and LLM-based pairwise rankers naturally capture cases where local inconsistencies and cycles occur. These effects, together with the soft–hard count mismatch discussed in Lemma 3.4, can introduce biases in pure Bradley-Terry models. Bayes-ECR addresses this through its temperature calibration and soft-gating mechanisms. Consequently, Bayes-ECR exhibits inherent robustness to non-transitive relationships in pairwise rankings, as indicated by the strong results in experiments with non-perfect rankers in Figure 2-4 and Table 1 in the main text.
>
> **Question 1 (Q1). Sensitivity analysis on regressor’s uncertainty estimates.**
>
> We have conducted an additional sensitivity analysis by replacing the Random Forest regressor’s variance estimates with ground-truth absolute or squared errors scaled by $s \in \{0.1, 1.0, 10.0\}$. The results (**Fig 17 in Appendix A6**) show that overestimating uncertainty (e.g., s = 10.0) has minimal impact, while underestimating it (e.g., s = 0.1) degrades performance due to excessive confidence in the regressor. In practice, if sufficient labeled data exist, the scale of $\sigma_\text{re}^2$ can be calibrated using a small validation set. Otherwise, conservative estimates are preferable.
>
> **Q2. Computational cost of the Newton updates.**
>
> Pure newton updates, i.e., running the pipeline without the pairwise ranking calls, take < 1 ms per query on an Intel i7-13700 CPU for a reference set size k ≤ 1000, which reasonably covers real-world applications in data-scarce domains.
>
> **Reference**
>
> [1] Wijaya, Kevin Tirta, et al. "Post Hoc Regression Refinement via Pairwise Rankings." arXiv preprint arXiv:2508.16495 (2025).

---

### Official Review · Reviewer_rqoX · 2025-11-01

**Soundness:** 2
**Presentation:** 2
**Contribution:** 3
**Rating:** 6
**Confidence:** 2

**Summary:**

This paper proposes BAYES-ECR (Bayesian Enhancement with Calibrated Rankings), a novel post-training Bayesian refinement framework for regression models using pairwise ranking information. The method integrates the Gaussian likelihood from a pretrained regressor with a Bradley-Terry ranking likelihood, forming a strictly log-concave posterior that enables efficient MAP optimization.
The authors identify a key failure mode in prior ranking-based refinements (e.g., RankRefine): when the number of reference items increases, the Bradley-Terry likelihood can dominate due to curvature mismatch, causing bias. BAYES-ECR corrects this through temperature calibration and accuracy-aware soft gating, controlling the relative contribution of ranking information.
Empirical results across 12 datasets (including molecular ADMET and tabular tasks) show that BAYES-ECR significantly improves predictive accuracy and robustness

**Strengths:**

- The paper is with fair conceptual clarity and novelty. It cleanly reformulates post-training regression enhancement as a Bayesian inference problem rather than heuristic fusion, providing a unified probabilistic interpretation that subsumes RankRefine as a special case.

- The theoretical grounding is solid. The analysis of rank-likelihood dominance and curvature scaling (Lemmas 3.4–3.7) is rigorous and insightful, pinpointing why Bradley–Terry models can fail at large k and how temperature calibration mitigates this.

- The overall evaluation is comprehensive. The paper evaluates multiple variants (MAP, MLE, MLE-Temp, MLE-GatedTemp) across molecular and non-molecular datasets, using both oracle and LLM rankers. Ablations, runtime, and robustness studies are thorough.

**Weaknesses:**

I'm not the expert of reco.sys, but I tried my best to understand the paper. Here are some general concerns based on my understanding:

1. Despite the clean Bayesian framing, much of the improvement stems from calibration heuristics rather than a fundamentally new probabilistic model. It seems that the method extends, rather than transforms, RankRefine’s core idea.

2. The proofs dominate the narrative. Figures are dense and sometimes too mathematical (Fig. 1–3). The paper could benefit from a small worked example demonstrating Bayesian updates visually. While technically sound, the exposition can feel overloaded: long derivations, dense notation reuse, and sometimes unclear transitions (e.g., from Lemma 3.7 to Corollary 3.8). Some results could move to Appendix without loss. Clearer clarification on the background will be appreciated.

**Questions:**

- How sensitive is BAYES-ECR to the regressor variance σ²_re? Does under- or overestimated uncertainty systematically bias the refinement?

- The analysis assumes symmetric flip noise. What happens if the ranker has systematic bias (e.g., consistently underestimates large values)? Could the Bayesian posterior incorporate a bias term instead of gating?

---

> ### Author Response · Authors · 2025-11-21
>
> We thank the reviewer for their insightful feedback. We have conducted additional experiments to address the reviewer’s questions. The results are presented in Appendix A6, which we reference in our detailed answers below.
>
> **Weakness 1 (W1). How does Bayes-ECR transform RankRefine’s core idea?**
>
> We fully agree with the reviewer that Bayes-ECR addresses the same goal as RankRefine: enhancing a pretrained regressor using pairwise rankings without retraining, but the similarity lies only in the question itself. Bayes-ECR introduces a novel, principled Bayesian fusion of the Gaussian regressor and Bradley-Terry rank likelihoods, replacing RankRefine’s heuristic weighting with a theoretically grounded inference framework supported by extensive analyses.
>
> **W1.2 Much of the improvement seems to stem from the calibration.**
>
> We tested this hypothesis and found out that, crucially, a major portion of the performance gain w.r.t. RankRefine stems from the Bayesian formulation rather than the calibration. When comparing the non-calibrated (NC) BAYES-ECR with RankRefine at reference set size k=20, NC BAYES-ECR (**Fig. 16 in Appendix A6**) consistently achieves lower error across datasets, often by a large margin. The results demonstrate that the Bayesian framework itself drives substantial improvement, while calibration enables BAYES-ECR to reach its full potential.
>
> **W2. Presentation and exposition can feel overloaded.**
>
> We appreciate the reviewer’s feedback and agree that improving accessibility will strengthen the paper. We will add an illustrative example in the camera-ready version to demonstrate the mechanism of Bayes-ECR. We will also rearrange parts of the mathematical exposition to better highlight the main ideas before diving into formal proofs.
>
> **Question 1 (Q1). Sensitivity analysis on regressor’s uncertainty estimates.**
>
> We have conducted an additional sensitivity analysis by replacing the Random Forest regressor’s variance estimates with ground-truth absolute or squared errors scaled by $s \in \{0.1, 1.0, 10.0 \}$. The results (**Fig. 17 in Appendix A6**) show that overestimating uncertainty (e.g., s = 10.0) has minimal impact, while underestimating it (e.g., s = 0.1) degrades performance due to excessive confidence in the regressor. Overconfidence could be particularly problematic in our small-data setting because the regressor must handle out-of-distribution samples, a challenging scenario for learning-based models. In practice, if sufficient labeled data exist, the scale of $\sigma_\text{re}^2$ can be calibrated using a small validation set. Otherwise, conservative estimates are preferable.
>
> **Q2.1. What happens if the ranker has a systematic bias (non-uniform flip noise).**
>
> We observe that Bayes-ECR is robust to non-uniform flip noise compared to both RankRefine and Projection baselines. As shown in Table 1 in the main text, Bayes-ECR outperforms both baselines when paired with two distinct LLM rankers (Claude 4 and ChatGPT 5-Thinking). Since LLMs are data-driven models, they inherently tend to exhibit non-uniform error patterns, particularly when handling out-of-distribution (OOD) samples versus in-distribution. Thus, the strong performance of Bayes-ECR with both LLM rankers suggests that Bayes-ECR is robust to non-uniform flip noise.
>
> **Q2.2. Can we use a bias term in the Bayesian posterior instead of gating?**
>
> In principle, the gating mechanism can be replaced (or complemented) by an explicit bias term within the Bayesian posterior.
> Specifically, in Eq. (4),
> $$\mathcal{L}(y) = -\frac{1}{2\sigma_\text{re}^2}(\hat{y}_0^\text{re} - y)^2 + \sum_i ^k[r_i \text{log} s(y-y_i) + (1 - r_i) \text{log}(1-s(y-y_i))] + \text{log} p(y),$$
>
> the Bradley-Terry terms $s(y-y_i)$ can be modified to $s(y-y_i+b)$, i.e.,
>
> $$\mathcal{L}(y) = -\frac{1}{2\sigma_\text{re}^2}(\hat{y}_0^\text{re} - y)^2 + \sum_i ^k[r_i \text{log} s(y-y_i+b) + (1 - r_i) \text{log}(1-s(y-y_i+b))] + \text{log} p(y),$$
> where b corrects for a systematic shift in the ranker logits. Estimating b accurately would likely require a small calibration on a hold-out set. We will discuss this in the paper and leave the full investigation  for future work.

---

### Meta-Review · Area_Chair_Cr8A · 2026-01-11

**Summary:**

The paper addresses the problem of post-training refinement of regression models. The Authors introduce a plug-and-play Bayesian framework for this purpose, based on pairwise ranking information, which is often easier to obtain than precise numeric labels or full rankings. The method combines the regressor’s predictive likelihood with a ranking-based Bradley–Terry likelihood to yield a well-behaved posterior that can be optimized efficiently. The Authors analyze theoretical properties of the approach, identify and correct a fundamental failure mode in ranking likelihoods via calibration, show that prior methods arise as special cases, and demonstrate substantial performance gains across multiple datasets.

The initial scores are relatively widely spread, ranging from 4 to 8. The main concerns relate to the presentation of the paper, the method being largely driven by a heuristic built on prior work, the assumption of conditional independence between the regressor and the ranker, potential inconsistencies in pairwise comparisons, the use of pairwise instead of listwise ranking, insufficiently explained steps of the proposed method, and a limited number of use cases/experimental evaluation. The Authors’ rebuttal clarifies the main issues and provides additional results, strengthening the paper and making it a solid contribution.

**Reviewer Concerns:**

The Authors successfully address the concerns, particularly those raised by the more critical reviewers, by clarifying the algorithm, discussing the advantages and limitations of pairwise versus listwise rankings, deriving an extension of the method to listwise rankings, and explaining that the approach has been validated on problems from additional domains.

**Reviewer Scores:**

- Reviewer rqoX would likely keep the score of 6 or slightly increase it
- Reviewer zBQs would likely keep the score of 8
- Reviewer JneQ would likely increase the score from 4 to 5 or 6
- Reviewer aXK6 would likely increase the score from 4 to 5 or 6

---

### Decision · Program_Chairs · 2026-01-26

Accept (Poster)